# Comparisons of Polarimetric Radio Occultation Measurements with WRF Model Simulation for Tropical Cyclones

Shu-Ya Chen[1], Ying-Hwa Kuo[2], Hsiu-Wen Li[1], Ramon Padullés[3], Estel Cardellach[3], Francis Joseph Turk[4]

[1]Global Atmospheric Observation and Data Applications Research Center, National Central University, Taoyuan City, Taiwan
[2]UCAR Community Programs, University Corporation for Atmospheric Research, Boulder, CO, USA
[3]Institute of Space Sciences (ICE-CSIC), Institute of Space Studies of Catalonia (IEEC), Boulder, CO, Spain
[4]Jet Propulsion Laboratory, California Institute of Technology, Pasadena, CA, USA

*Correspondence to*: Shu-Ya Chen (shuyachen@ncu.edu.tw)

**Abstract.** A novel radio occultation (RO) technique, polarimetric RO (PRO), has recently been developed to measure differential polarimetric phase shift together with traditional RO products such as bending angle and refractivity. PRO observations have been shown to be associated with the vertical structure of cloud hydrometeors. With this unique measurement capability, the PRO soundings could potentially be used to evaluate model microphysics parameterization. This study compared PRO observations with WRF simulations of three typhoon cases in 2019 and 2021, initialized with ERA5 and NCEP FNL global analysis, respectively, with five microphysics parameterizations (Purdue Lin, WSM6, Goddard, Thompson, and Morrison). There is notable variability in the distribution of the model's hydrometeors, which could be affected by the initial conditions, microphysics parameterization schemes, typhoon locations, circulation, and rainbands. The results in this study show that WRF simulations using the Goddard, Thompson, and Morrison schemes generally place the peak differential phase at the altitudes close to those observed by PAZ PRO. Among them, the Goddard microphysics scheme performs best in typhoon track prediction and the simulation of maximum differential phase shift when compared with PRO observations. The ensemble mean from 36 ensemble forecasts also exhibits consistent results with the deterministic run. The comparative results demonstrate that PRO data have the potential to evaluate the performance of different microphysics schemes in numerical models.

## 1 Introduction

The Global Navigation Satellite System (GNSS) radio occultation (RO) technique detects the phase delay induced by the atmospheric variation of air density and water vapor. It has been demonstrated that RO observations are useful for climate monitoring and numerical weather prediction, e.g., Anthes (2011), Gleisner et al. (2020, 2022), and Ho et al. (2020). The RO data provide profiles of atmospheric bending angle and refractivity with high accuracy and precision, as well as global coverage. These RO products can assist in understanding the atmospheric thermodynamic process (e.g., Chen et al., 2020, 2021; Chang

and Yang, 2022; Hong et al., 2023), and the data have been routinely assimilated for operational weather prediction (e.g., Cucurull, 2023; Lien et al., 2021; Ruston and Healy, 2021). In recent years, a novel technique known as polarimetric radio occultation (PRO) has been introduced (Cardellach et al., 2015). This technique employs a dual-polarization RO receiver, which acquires GNSS signals in both horizontal and vertical polarizations. The polarized horizontal and vertical signals passing through atmospheric hydrometeors result in different phase delays due to the shape of hydrometeors. By analyzing the differential polarimetric phase shift (i.e., phase delay of the horizontally polarized signal with respect to the vertically polarized one), we can gain a better understanding of the structure and composition of cloud hydrometeors. The PRO data can provide not only traditional atmospheric thermodynamic profiles of temperature and moisture but also relevant information about hydrometeors in the cloud (Cardellach et al., 2018).

The Spanish PAZ satellite was successfully launched in February 2018 and began operations in May of the same year (Cardellach et al., 2019). Since then, the PAZ Radio Occultations through Heavy Precipitation (ROHP) receiver payload has provided about 200 PRO soundings daily around the globe. Starting in 2023, more PRO data have been available from the commercial company Spire Global (Talpe et al., 2025; Padullés et al., 2024), which has acquired over 2,000 PRO profiles per day during some periods. With abundant PRO observations, it starts to be feasible to envisage PRO applications to monitoring the atmospheric environment, for use in numerical models, and to better characterize the link between thermodynamics and intense precipitation in the atmosphere (Turk et al., 2024). Several studies have substantiated the PAZ satellite's capability of sensing precipitation. Cardellach et al. (2019) and Padullés et al. (2020) validated the PRO data against the joint NASA/JAXA Global Precipitation Measurement (GPM) mission, Integrated Multi-satellitE Retrievals for GPM (IMERG) global precipitation dataset, showing a high correlation between the differential phase shift from PAZ PRO and satellite measurements of precipitation, not only in the lower troposphere but also for frozen particles above the freezing level. Padullés et al. (2022) complemented the validation with data from the GPM Microwave Imager (GMI), the GPM Dual Frequency Precipitation Radar (DPR) and the W-band Cloudsat radar, demonstrating the sensitivity to the vertical extent of the ice hydrometeors above the freezing level. Murphy et al. (2019) used airborne PRO data to compare with numerical simulations for an intense atmospheric river event. The simulated differential polarimetric phase shifts from the Weather Research and Forecasting (WRF) model with two model microphysics schemes showed significant differences, with differences larger than the PRO noise level (thus detectable by this technique). These results suggest the potential of using PRO data for validating model microphysics representation.

With climate change, severe weather events, such as intense typhoons, accompanied by extreme precipitation have been increasing (Tabari, 2020; Masson-Delmotte et al., 2021), which demands more accurate precipitation forecast. However, the prediction of precipitation is associated with the cloud microphysical parameterization of a numerical model. Currently, the microphysical processes in most of the weather and climate models are represented as bulk microphysics parameterizations without a lot of details (such as the hydrometeor size distribution); therefore, the evaluation of these parameterizations has been a challenge. Hristova-Veleva et al. (2021) evaluated WRF simulations with different microphysical parameters against multi-parameter satellite data and their analyses revealed significant differences that highlighted the uncertainty in model

microphysics parameterization. This points to the need for more observations to evaluate the cloud microphysical parameterization and to gain further insights on the microphysical processes.

There are a few types of satellite remote sensing observations for precipitation. Satellite radiance measurements can detect atmospheric precipitation with different water vapor channels, but they are limited by vertical resolution and the cloud area coverage. The PRO observations offer a possibility for evaluating the performance of various cloud microphysics schemes.

The motivation of this study is to assess *whether the PRO observation can be used to evaluate the performance of model cloud microphysical parameterizations on Typhoon cases.*

The study focuses on three typhoon cases, including two in 2019 and one in 2021. The first one is Typhoon Bualoi, which formed on 19 October 2019, and under a favorable condition of low vertical wind shear and warm sea surface temperature. It rapidly intensified to become a typhoon on the next day. Two days later, it intensified rapidly into a Category-5 typhoon. Even

though Bualoi did not make landfall in Japan, the heavy rainfall caused floods and landslides, which resulted in casualties and substantial property damage. The second case is Typhoon Matmo, which formed on 28 October 2019 over the South China Sea and then weakened after making landfall in central Vietnam on 30 October. It brought strong winds and heavy rainfall, causing flooding and road closures. Typhoon Matmo destroyed ~2,700 houses, causing more than 165 million USD in damage in Vietnam. The third case is Typhoon Kompasu in 2021, which formed through the merging of two tropical depressions

embedded within a monsoonal circulation. The typhoon affected many areas, including the Philippines, Taiwan, southeast China, and Vietnam. It caused significant damage to the Philippines, and consequently, the name "Kompasu" was removed from the naming list of tropical cyclones. These three typhoon cases were selected because at least one PRO profile was located very close to the vertex center for each case during the maturity stage of the typhoons. The relative locations can be found in Fig. 1 and Table 1.

**Table 1. A list of observations near each typhoon case, including GNSS RO from FORMOSAT-7/COSMIC-2 (wetPf2), PRO from PAZ (PAZ1), and radiosonde (sonPrf).**

| Typhoon | Simulation Period | Observations | location |
|---|---|---|---|
| BUALOI | 2019/10/23 06 UTC ~ 2019/10/24 00 UTC | PAZ1.2019.296.21.41.G14 | 26.340ºN, 141.570ºE |
| | | wetPf2_C2E1.2019.296.21.25.R17 | 23.215ºN, 137.081ºE |
| MATMO | 2019/10/29 18 UTC ~ 2019/10/30 12 UTC | PAZ1.2019.303.09.35.G16 | 14.370ºN, 109.300ºE |
| | | wetPf2_C2E2.2019.303.07.09.G05 | 13.070ºN, 106.917ºE |
| | | wetPf2_C2E2.2019.303.07.22.G27 | 16.720ºN, 106.822ºE |
| | | wetPf2_C2E2.2019.303.09.07.R21 | 16.549ºN, 105.140ºE |
| | | wetPf2_C2E3.2019.303.09.33.G15 | 9.495ºN, 113.326ºE |
| | | wetPf2_C2E4.2019.303.07.39.G27 | 12.254ºN, 105.236ºE |
| | | wetPf2_C2E4.2019.303.09.13.G15 | 15.483ºN, 111.470ºE |
| | | wetPf2_C2E4.2019.303.11.09.R07 | 13.757ºN, 109.022ºE |
| | | wetPf2_C2E4.2019.303.11.10.G03 | 10.255ºN, 107.977ºE |
| | | wetPf2_C2E5.2019.303.10.17.G22 | 13.532ºN, 111.725ºE |
| KOMPASU | 2021/10/12 06 UTC ~ 2021/10/13 00 UTC | PAZ1.2021.285.23.27.G04 | 18.840ºN, 112.650ºE |
| | | wetPf2_C2E2.2021.285.21.37.G24 | 22.326ºN, 114.932ºE |
| | | wetPf2_C2E5.2021.285.22.03.G03 | 15.476ºN, 111.449ºE |
| | | wetPf2_C2E5.2021.285.22.06.R02 | 21.835ºN, 117.528ºE |
| | | wetPf2_C2E5.2021.285.22.12.G24 | 15.776ºN, 111.701ºE |
| | | sonPrf_C2E2.2021.285.21.37.G24 | 22.320ºN, 114.170ºE |

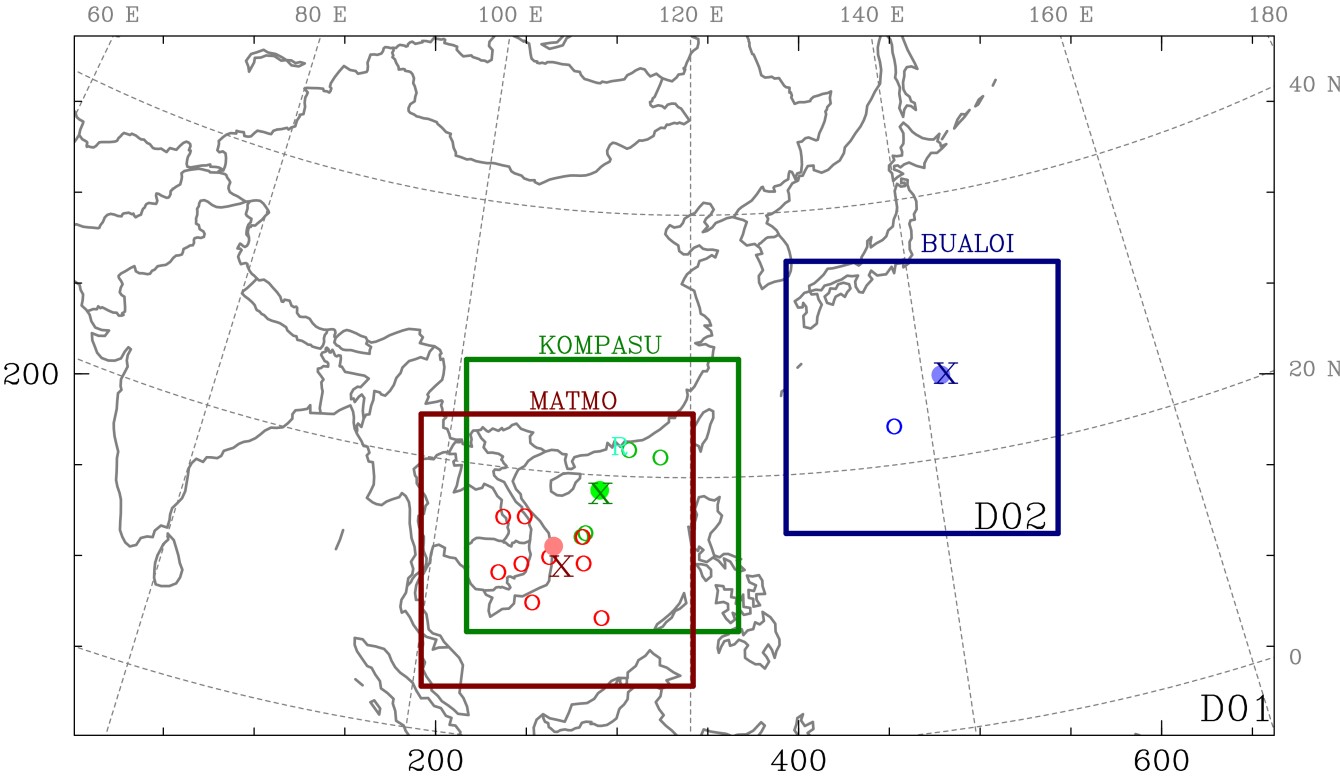

**Figure 1. The domain coverage with the same Domain 1 (D01) and varied Domain 2 (D02) for individual cases, as indicated by colored squares. The symbols represent different data sources: solid dots for PRO data from PAZ, hollow circles for GNSS RO data, the character 'R' for radiosonde, and cross signs for the JTWC typhoon locations at the ending forecast time of each simulation case.**

In this study, we perform high-resolution WRF model simulations for the three typhoon cases and compare the simulated differential polarimetric phase shifts with the PAZ PRO observations. The WRF model configuration and the experimental design are described in Sect. 2, and the PRO forward operator is introduced in the same section. Sect. 3 presents the simulated results with different initial conditions and microphysical parameterizations. Verification against the PRO observations for the three typhoon cases are presented in Sect. 4, and the analysis of the uncertainties caused by various factors, including model initial conditions, difference between model and observed storm location, as well as details of the simulated cloud distributions, etc., are discussed in the same section. Finally, the summary and conclusion are presented in Sect. 5.

## 2 WRF model and the PRO forward operator

### 2.1 Model configuration and experimental designs

The Advanced Research version of WRF (WRF-ARW, hereinafter the WRF Model; Skamarock et al., 2021) is a fully compressible, nonhydrostatic model widely used by the research community and operational centers. In this study, the WRF

model version 4.2 is used with two model domains at resolutions of 15 km and 3 km, respectively (Fig. 1). The outermost domain (D01) is fixed for all three typhoon cases with 662×386 grid cells, and the inner domain (D02), with 751×751 grid cells, is placed near the center of the storm for each individual typhoon case. The domain coverages for each typhoon case are depicted in Fig. 1. The model consists of 52 layers with the model top at 20 hPa. Each simulation begins with a cold start and is integrated for 18 hours to spin up the model microphysics. The initial time for the WRF simulation varies for each typhoon case, with initial times set at 0600 UTC 23 October 2019 for Typhoon Bualoi, 1800 UTC 29 October 2019 for Typhoon Matmo, and 0600 UTC 12 October 2021 for Typhoon Kompasu (Table 1).

The accuracy of model simulations is impacted by uncertainties associated with initial conditions and model formulation such as physical parameterization (Ehrendorfer, 1997). To evaluate the uncertainties, we conduct WRF simulations with two sets of initial conditions and five microphysics parameterization schemes. Two global analyses, including NCEP FNL (Final) analysis and ERA5 reanalysis, were adopted for the WRF initial conditions, while NCEP FNL at 6-h interval and ERA5 at 1-h interval were used for the boundary conditions. The two initial conditions were used to assess whether uncertainties related to microphysics schemes persist across different initial states. For both initial conditions, the same horizontal resolution of $0.25^{o}$ by $0.25^{o}$ is adopted. Since the focus of this study is cloud microphysical parameterization, we keep all other physical parameterizations the same for all experiments, with the exception of microphysics. These common physical parameterizations include: Kain–Fritsch cumulus parameterization (Kain, 2004), Yonsei University (YSU) planetary boundary layer (PBL) parameterization scheme (Hong et al., 2006), and a new version of the Rapid Radiative Transfer Model (RRTMG) for the radiation effects (Iacono et al., 2008). Notice that the cumulus convective parameterization scheme was applied only for the 15-km domain.

It should be noted that only relative humidity from either ERA5 or NCEP FNL is ingested in the WRF preprocessing system (WPS) for the model initialization. None of the hydrometeor information from the global analyses was used in the WRF initial condition. The precipitation structure (i.e., hydrometeor distribution) in the model has to be developed through model integration with the microphysics parameterization. There are many options of microphysics schemes available in the WRF model, and each parameterization has its own unique way of handling microphysical processes. In this study, we evaluate the performance of five microphysics schemes, such as Purdue Lin scheme (Chen & Sun, 2002), WSM6 6-class graupel scheme (Hong and Lim 2006), Goddard 4-ice scheme (Tao et al., 1989, 2016), Thompson graupel scheme (Thompson et al., 2008), and Morrison 2-moment scheme (Morrison et al., 2009). Table 2 lists the abbreviated names and corresponding WRF options for these microphysics schemes used in the study. These schemes have been used in operations and research by the Numerical Weather Prediction (NWP) community.

Table 2. The abbreviated names and microphysics schemes used in the study and their corresponding WRF options.

| Abbreviated name | Microphysics | WRF options |
|---|---|---|
| PurdueLin | Purdue Lin scheme | 2 |
| WSM6 | WRF Single-Moment 6-class scheme | 6 |
| Goddard | Goddard 4-ice microphysics scheme | 7 |
| Thompson | New Thompson et al. scheme | 8 |

| Morrison | Morrison double-moment scheme | 10 |
|----------|-------------------------------|-----|

The microphysics parameterization schemes model the microphysical processes and the evolution of different hydrometeor species, such as water vapor, cloud water, rain water, ice, snow, graupel, hail, etc. Within these five schemes, Thompson and Morrison schemes both are a double moment scheme, which consider the number concentration and cloud condensation nuclei effects. The Purdue Lin, WSM6, and Goddard schemes are single moment schemes, and all with ice, snow, and graupel processes. The difference between Goddard scheme and the other two is that it predicts hail and graupel separately, which provides effective radii for radiation. More detail for the microphysics schemes in the WRF model can be found in Skamarock et al. (2021).

WRF simulations initialized with the ERA5 and NCEP FNL and the five microphysics schemes are run for 18 h for the three typhoon cases. Thus, there are a total of 30 WRF simulations. We identify each run by a string composed of the initial data source, the typhoon case name, and the microphysics scheme. For example, the simulation initialized from ERA5 and used the Purdue Lin scheme for typhoon Bualoi is named ERA5_Bualoi_PurdueLin.

## 2.2 PRO forward model

The variable from polarimetric RO is the differential phase shift ($\Delta\phi$) between the horizontal wave ($\phi_h$) and vertical wave ($\phi_v$), which is the additional excess phase delay due to precipitation (Cardellach et al., 2015). The polarimetric phase shift in the unit of mm can be represented as

$$\Delta\phi = \int_L K_{dp}(l)dl \tag{1}$$

where $K_{dp}$ is the specific differential phase in mm km⁻¹, and $L$ is the path length of the radio link. Since the $K_{dp}$ is induced by the difference in scattering properties of hydrometeor particles, a simple linear relation between the water content (WC) and $K_{dp}$ is adopted, following the formula presented in Bringi and Chandrasekar (2001) and Padullés et al. (2022), as below.

$$K_{dp}(WC) = \frac{1}{2}C\rho \times WC \times (1 - ar) \tag{2}$$

where WC indicates the water content of any hydrometeors in ice, snow, rain, etc. in units of g cm⁻³, $\rho$ is the particle density in units of g cm⁻³, and $ar$ is the assumed dimensionless axis ratio of the particle. $C$ is the Rayleigh scattering at the GPS frequency, which is a proportionality constant of 1.6 (g cm⁻³)⁻². Considering the variables ($\rho$ and $ar$) vary for each hydrometeor, we followed the function presented in Fig. 9 of Padullés et al. (2022), which provided a profile for $\rho \times (1 - ar)$ from the freezing level to the cloud top, to calculate the specific differential phase for this study. We create a lookup table based on the profile for calculating $K_{dp}$. The function is used only for frozen hydrometeors, and the coefficients at heights below the freezing level and above the cloud top are set to the same constant values as those at these two levels. Regarding liquid hydrometeors, such as rain, we adopt $\rho$ and $ar$ as 1 g cm⁻³ and 0.95, respectively, as suggested by Chang et al. (2009) for liquid rain during typhoon events. Thus, a fixed value of $\rho \times (1 - ar) = 0.05$ is used for rain. Even though we consider variations and types of hydrometeors, the amounts and effective axis ratios, which are determined by particle size distribution and orientation, are still challenging to evaluate, thus introducing a limitation to the approach.

To calculate the simulated differential phase shift, the WC on each grid point was converted from the three-dimensional
hydrometeors into mixing ratio, which included rain, ice, snow, graupel, hail, clouds, etc. Since each PAZ PRO sounding
provides latitude, longitude, and height along the 220 ray paths (Padullés et al., 2024), the WRF simulations were interpolated
accordingly. Then, the simulated $K_{dp}$ was calculated by Equation (2) at the same location and height as PAZ PRO, and the
integral of $K_{dp}$ along the raypath through the Equation (1) for the $\Delta\phi$ can be derived. A time interpolation by selecting two
model outputs closest to the PRO observation time was conducted as well for the comparisons.

## 3 Simulated results

The model initializations from these two data sources, NCEP FNL and ERA5, already showed significant differences
before the model forward integration. Using Typhoon Bualoi as an example the initial intensity (i.e. sea level pressure, SLP)
and total precipitable water (TPW) from NCEP FNL (Fig. 2b) is more intense than that from ERA5 (Fig. 2a). The difference
in intensities between the two analyses can be more than 30 hPa. Besides, the TPW distribution initialized with NCEP is about
175 80 mm near the inner core and exhibits more symmetry than that of ERA5. This discrepancy illustrates that the model
initialization from different global analyses already possesses the uncertainties from the beginning.

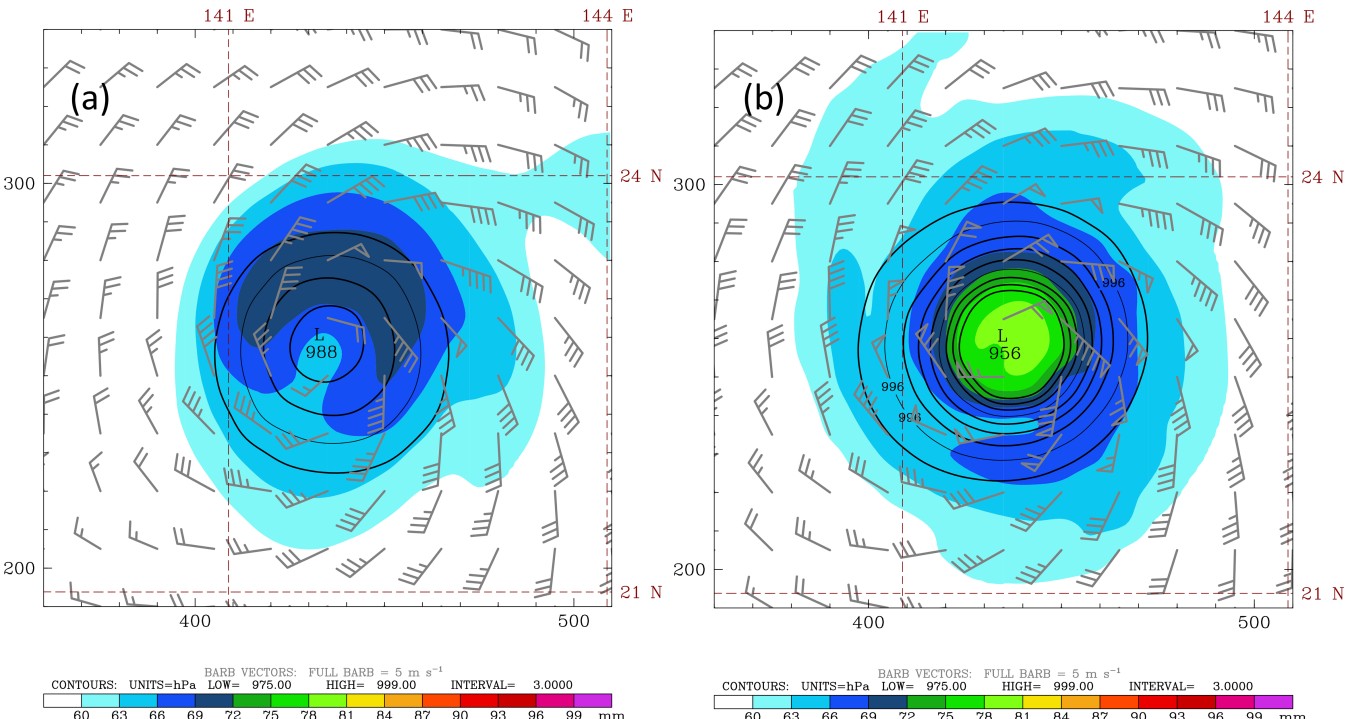

**Figure 2. Total precipitable water in color (mm), sea level pressure in contour (hPa), and wind vectors (m s⁻¹) for the WRF initial conditions from (a) ERA5 and (b) NCEP FNL for the Bualoi case.**

Besides the initial sources, the model parameterizations could also result in variability. Even if the initial condition is identical, using different microphysics parameterization schemes show distinct patterns in precipitation. Fig. 3 displays the SLP and TPW at 18-h forecast from two initial conditions with five microphysics schemes for Bualoi. The simulations show that all the simulated typhoon vortices have a location error with a westward shift compared to the best track. The maximum location error at the 18-h forecast is about 100 km. After the short-term forecast, the simulated typhoon intensity initialized

with ERA5 still exhibited a weaker vortex than that from NCEP. Despite a relatively weaker and drier circulation from the ERA5 initial condition, the TPW in the typhoon circulation is increased to near 100 mm through the WRF integration (Fig. 3a-e). Moreover, the maximum TPW for WRF runs with WSM6 and Morrison microphysics initialized with ERA5 (i.e., ERA5_Bualoi_WSM6 and ERA5_Bualoi_Morrison, Fig. 3b,e) are larger than that initialized with NCEP FNL (Fig. 3g,j). The horizontal distributions of TPW using the Goddard and Thompson schemes (Fig. 3c,d,h,i) tend to be broader than the other

schemes, regardless of the global analysis used to initialize the model. Generally, the simulations show an intense TPW over the western or northwestern part of the vortex, and some differences in scattering precipitation distribution due to typhoon rotation. Fig. 3 highlights a large variability in the TPW's distributions with different microphysics schemes, which illustrates the uncertainty of the parameterization schemes.

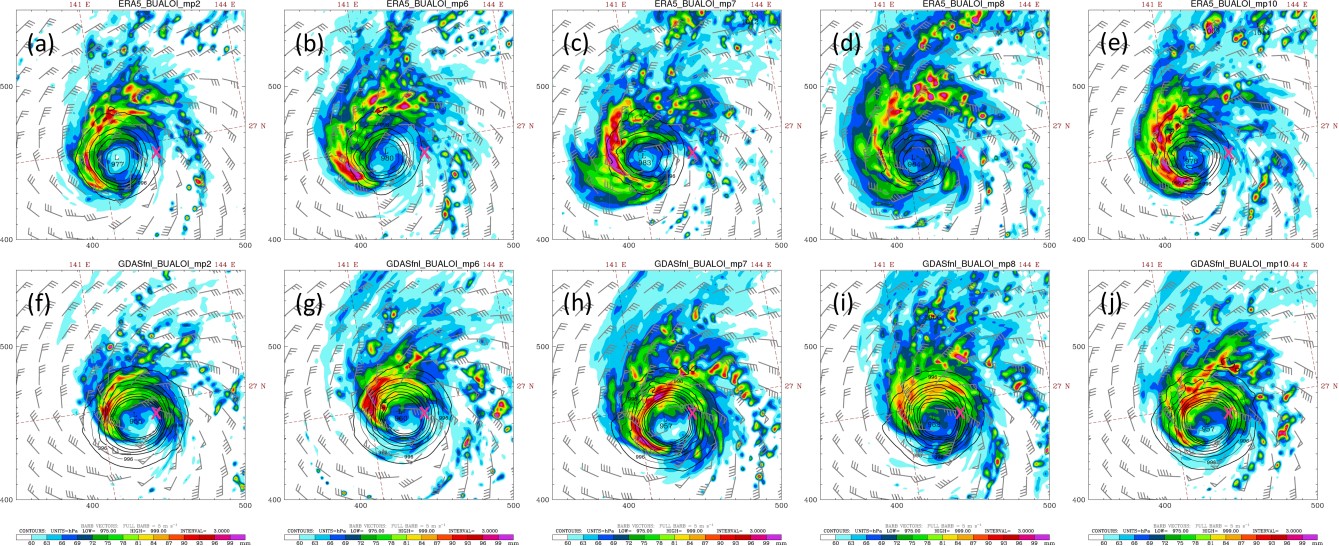

**Figure 3.** Panels (a)–(e) are similar to Figure 2a for total precipitable water (in color), sea level pressure (in contour), and wind vectors with the initial condition from ERA5, but for WRF 18-hour forecast with different microphysics schemes: (a) Purdue Lin scheme, (b) WSM 6-class graupel scheme, (c) Goddard 4-ice scheme, (d) Thompson graupel scheme, and (e) Morrison 2-moment scheme. Panels (f)-(j) are the same as (a)-(e), but for the initial conditions from NCEP FNL. The red cross sign in each panel, at 26.8ºN and 142.1ºE indicates the observed location of Typhoon Bualoi from the JTWC's best track.

To evaluate the performance of experiments with different microphysics and initial conditions, the simulated tracks from all cases are compared with the best track from the Joint Typhoon Warning Center (JTWC). Under the same initial condition (either NCEP FNL or ERA5), the simulated typhoon tracks using different microphysics schemes exhibit high similarity, indicating a relatively weak sensitivity of track prediction to microphysics schemes (Fig. 4a–c). However, the use of different

initial conditions leads to a clear bifurcation in the track patterns, forming two distinct groups corresponding to the two initial

datasets, except for ERA5_Matmo_WSM6, which shows a significantly deviated track compared to the others (Fig. 4b). The analysis of track and intensity errors reveals that most simulated tracks have errors of less than 100 km during the 18-hour forecasts. The simulated intensity generally deviates by less than 12 hPa, except for the Bualoi cases, which exhibit stronger development and larger errors (figure not shown). Figures 4d and 4e show the mean track and intensity errors, respectively, averaged across the three typhoon cases. Among all the microphysics schemes, the Goddard scheme generally performs better

with smaller track errors regardless of the initial condition used (Fig. 4d). However, it exhibits a larger mean intensity error in simulations initialized with ERA5 (Fig. 4e). When comparing track and intensity performance for the three cases individually, no consistent pattern emerges to indicate that any specific combination of initial condition and microphysics scheme outperforms the others across all metrics. To highlight the effect of microphysics, the subsequent discussion will focus on experiments initialized with the ERA5.

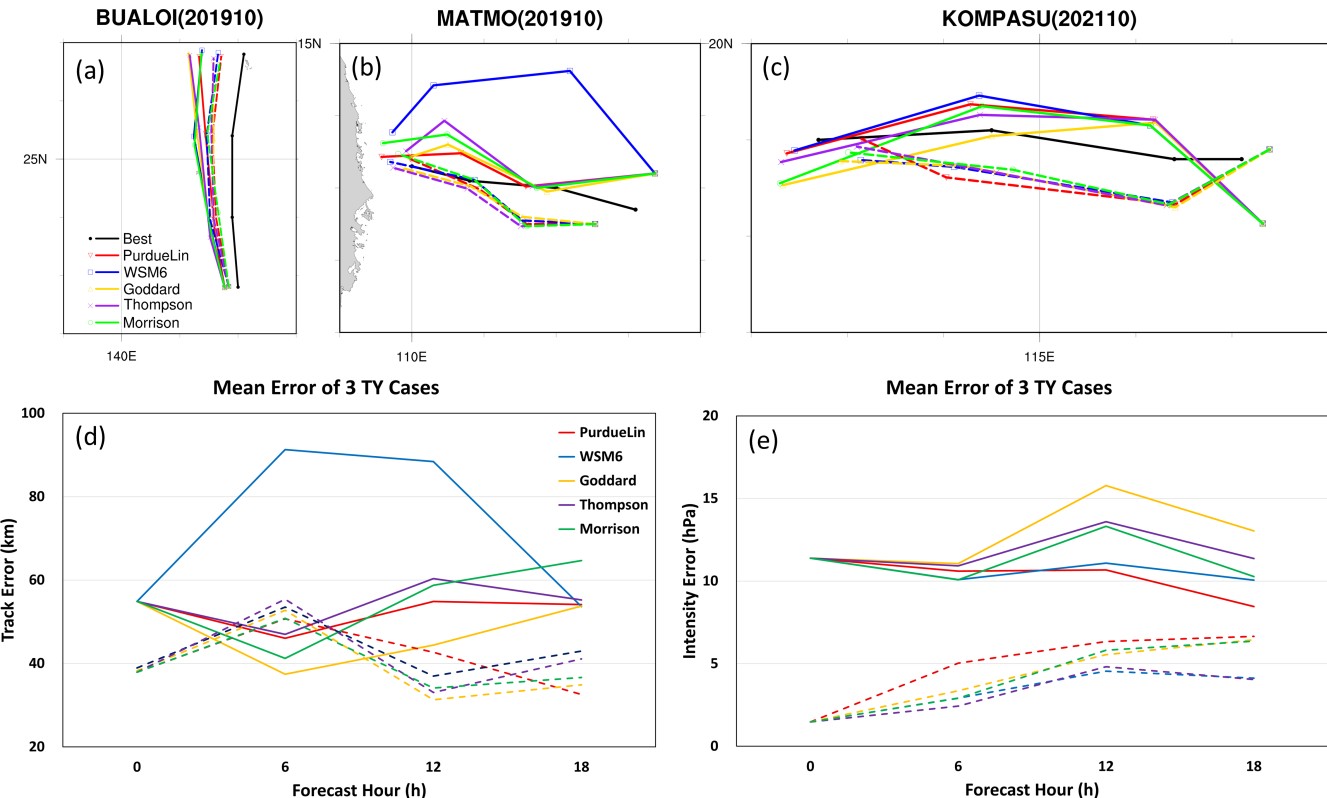

**Figure 4. Simulated tracks over time for Typhoons (a) Bualoi, (b) Matmo, and (c) Kompasu. The best track from JTWC is shown as a black line. Panels (d) and (e) show the averaged track errors and intensity errors, respectively, from all three typhoon cases. Solid lines represent simulations with ERA5 initial conditions, while dashed lines represent those with NCEP FNL initial conditions. Lines in different colors represent different microphysics schemes.**

Since traditional GNSS RO data have high accuracy and high vertical resolution, they can be used as verification data. For the three typhoon cases, there are a few soundings close to the typhoon vortex within 3h time window and 5° latitude-

longitude radius, including one traditional RO for Typhoon Bualoi, nine ROs for Typhoon Matmo, four ROs and one radiosonde for Typhoon Kompasu. Therefore, a total of 15 observations can be used for the sounding verification (Table 1). They are indicated in Fig. 1 with the typhoon locations from the JTWC best track. The WRF forecasts, at 1-h intervals, are interpolated to the specific times and locations of 15 soundings for comparison. Fig. 5 shows the temperature and water vapor mixing ratio verification. It is evident that the ERA5_Morrison (green curve) has less error in temperature than the other schemes above 10 km (Fig. 5a). However, it is not distinguishable in the water vapor mixing ratio (Fig. 5b). The overall vertical mean error for the five microphysics ranges from -0.34 ~ -0.42 °C in temperature, and -0.16 ~ -0.18 g kg$^{-1}$ in moisture. To avoid incorporating a priori information due to the 1DVAR retrieval for GNSS RO data, refractivity verification is also conducted (figure not shown). It exhibits a variation pattern consistent with the moisture verification as in Fig. 5b. In general, the environmental verifications against the traditional RO and radiosonde do not show significant difference among the five microphysics schemes (Fig. 5). WRF simulations with different initial conditions and microphysical parameterizations can produce significant difference in cloud, precipitation structure, and hydrometeors distributions. However, their impact on the large-scale variables is relatively small for an 18-h forecast.

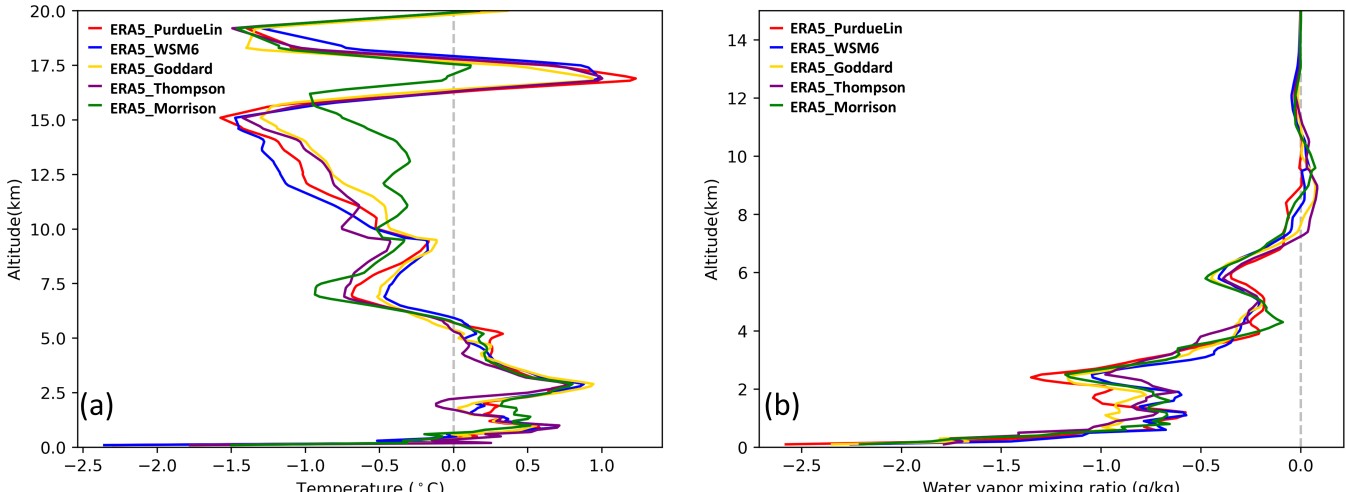

**Figure 5. The mean differences for verifications against soundings (14 GNSS RO and 1 radiosonde) in (a) temperature and (b) water vapor mixing ratio across all simulations. The red curve represents the PurdueLin scheme, blue represents WSM6, yellow represents Goddard, purple represents Thompson, and green represents the Morrison scheme.**

## 4 Verification against PRO observations

The PRO profile, sensitive to precipitation structure, could be used to evaluate the model hydrometeors' vertical structure in the sum of rain, ice, snow, graupel, hail, etc. We have one PAZ PRO profile available for each typhoon case (Table 1), and each of the PRO sounding is close to the individual typhoon center (Fig. 1); for example, the time and location differences between the PRO location and the storm center are 33 minute and 18.6 km, respectively, for Typhoon Kompasu. In addition, the WRF model has been integrated for more than 15 h before the comparison with the corresponding PRO data, allowing the

model to develop hydrometeors through the cloud microphysical process. To reduce the spatiotemporal error, two model outputs at time close to the PRO observation were linearly interpolated to the observation time of the PRO profile. Even with the time interpolation, the location of the simulated vortex could differ from the observed location as shown in Fig. 3, and the precipitation distribution would be shifted. To minimize the error due to location difference, the simulated storm center is relocated to the observed location. The ray trajectory is determined based on the relocated WRF simulations. Then, the

differential phase shift between the PAZ PRO and the WRF simulations can be compared. In addition, considering that cloud structures produced by different microphysics schemes can differ significantly in typhoon simulations, and that the simulated typhoon's asymmetric structure may also deviate from reality, two supplementary ray paths offset by ±0.5° (depending on the ray orientation) are included alongside the primary relocated ray to better account for small-scale structural discrepancies. The phase shift delays ($\Delta\phi$) along all three ray paths are averaged to yield a representative value for comparison with the PRO

observations. This approach helps mitigate the impact of misalignments between the observed and simulated storm structures, particularly in highly asymmetric systems such as typhoons. Moreover, the nature of PRO measurements represents an along-path integrated signal over a finite atmospheric volume rather than a single-point observation (Cardellach et al., 2024). This consideration ensures a more meaningful comparison between simulations and observations.

## 4.1 Deterministic run

Fig. 6 shows the WRF simulations with the ERA5 initial conditions and different microphysics for Typhoon Bualoi. The distribution of simulated hydrometeors along the ray path shows the rain species dominates below 5 km and a mixing of frozen precipitation above this height. Generally, the hydrometeors from the model show large variability. The PurdueLin and WSM6 schemes show a concentration of hydrometeors near the perigee point (26.34$^o$N, 141.57$^o$E), and they are both composed of larger graupel at 3-12 km (Fig. 6a,b). For the Goddard, Thompson, and Morrison microphysics, the snow dominates and has a

larger horizontal extent (Fig. 6c,d,e). Also, some hail present around 4-8 km for the Goddard scheme (ERA5_Bualoi_Goddard). The PAZ PRO observation shows a maximum differential phase shift of 35.1 mm at the height of 7.5 km for Typhoon Bualoi. The WRF simulations with various microphysics generally place the peak differential phase at an altitude close to the PAZ PRO observation, but with a large discrepancy in value. In particular, the PurdueLin scheme tends to underestimate frozen hydrometeors, resulting in the significant deviation from the observation, regardless of the initial conditions. The maximum

values of $\Delta\phi$ simulated by the PurdueLin, WSM6, Goddard, Thompson, and Morrison microphysics schemes are 10.6, 27.6, 35.9, 37.4, and 30.6 mm, respectively. Among these, the Goddard schemes and the double-moment schemes (Thompson and Morrison) show a peak $\Delta\phi$ at 5-8 km and large values, which are in agreement with the observation (Fig. 6c-e). Among the frozen hydrometeor species, the snow contributed the most, which agrees with the finding in Padullés et al. (2024). The vertical distribution of the simulated differential phase agrees better with the PRO data when using the Goddard scheme or the double

moment schemes. The best fitting is the experiment using the initial condition from ERA5 and the Goddard scheme, which produces a simulated maximum $\Delta\phi$ most closely aligned with that observed by PRO PAZ for the Bualoi case.

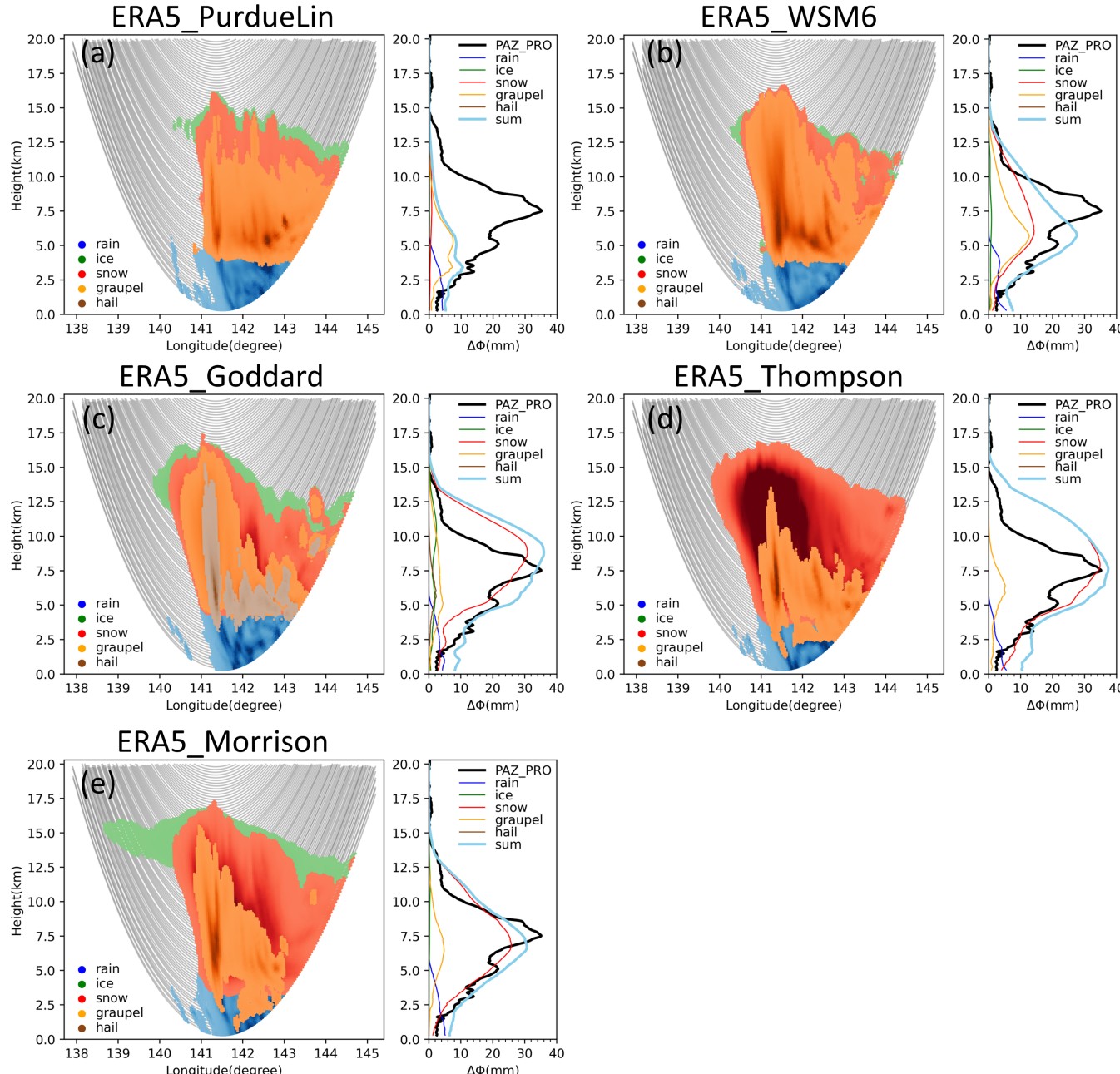

**Figure 6. Distribution of simulated hydrometeors along the PAZ raypath (gray curves) on the left panel, and the differential phase shift from PAZ observation (black curve), along with the calculated phase shift for each simulated hydrometeor species on the right panel for the Bualoi simulation with ERA5 as the model initial condition. The 15-h and 16-h WRF simulations are interpolated based on the PRO's time and location. The light blue curve represents the sum of all simulated phase shifts. Panels (a) to (e) represent different microphysics schemes for (a) PurdueLin, (b) WSM6, (c) Goddard, (d) Thompson, and (e) Morrison schemes.**

Fig. 7a shows the infrared satellite image of Typhoon Bualoi at 2230 UTC on 23 Oct. 2019, and the observed cloud-top temperature (CTT) is about -70°C. The simulated CTTs at the 16-hour forecast, using different microphysics schemes, are

285 shown in Fig. 7b-f. Generally, the extensive cloud coverage of typhoon Bualoi is well captured by the ERA5_Bualoi_Gaddard simulation, including the small-scale cloud clusters located to the south of the typhoon. In contrast, the simulations with the other microphysics schemes show large variations, e.g., a small coverage of low cloud-top temperature near the typhoon center for WSM6 (Fig. 7c), a loose typhoon structure for PurdueLin and Thompson (Fig. 7b,e), or an overly large typhoon circulation structure with Morrison (Fig. 7f), all that show significant discrepancies from the observations. The simulation with Goddard

(Fig. 7d) shows a solid vortex with a low cloud-top temperature of -70$^{\circ}$C, and the distribution aligns well with the infrared satellite image (Fig. 7a). Based on the comparisons of typhoon track, satellite image, and PRO observation (Fig. 4,6,7), the configuration using the ERA5 initial condition with the Goddard microphysics scheme will be adopted for the analysis of the other two typhoon cases.

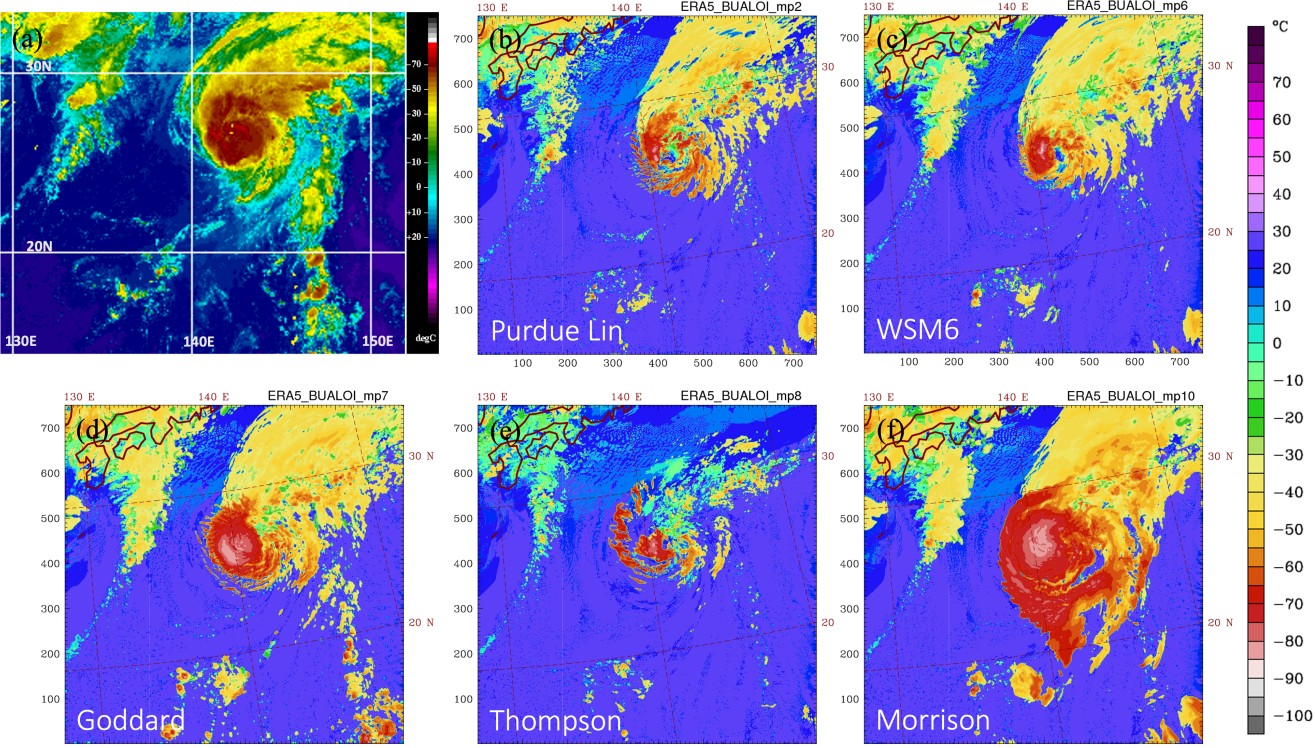

**Figure 7. (a) The infrared satellite image with NHC enhancement, adopted from the Cooperative Institute for Meteorological Satellite Studies/University of Wisconsin-Madison (CIMSS), for Typhoon Bualoi at 2230 UTC on 23 Oct. 2019. (b)-(f) The 16-h forecast (i.e., 2200 UTC 23 Oct.) cloud top temperature for typhoon Bualoi with the PurdueLin, WSM6, Goddard, Thompson, and Morrison microphysics, respectively, and the initial condition from ERA5.**

For Typhoon Matmo (2019), the PAZ PRO observation shows a large $\Delta\phi$ (more than 20 mm) extending from 3 km to 8

300 km with a maximum of 36.7 mm around 5 km (Fig. 8a). The WRF simulation, ERA5_Matmo_Goddard, also shows a large $\Delta\phi$ below 8 km, which is primarily contributed by snow in the mid-troposphere and by rain in the lower troposphere. In general, the simulated $\Delta\phi$ shows comparable values and vertical variation with the PRO measurement below 5km, although an

overestimation is found around 5-8 km. The simulated hydrometeors show that the graupel and hail occur mostly in the eastern part of the ray, and snow can be found along the ray path extending from 2.5 km to 12.5 km, with ice at higher altitudes. For Typhoon Kompasu, the PRO shows a $\Delta\phi$ of 20.9 mm at a higher altitude (8 km) and decreases below the height until 5 km, and then increases again toward the surface (Fig. 8b). The simulated $\Delta\phi$ does not reproduce such variation. The WRF simulation shows a large $\Delta\phi_{sum}$ of more than 35 mm at 6 km with an overestimation below this altitude compared to the observation. Notice that the verifications against ERA5 for ERA5_Kompasu_Goddard show larger RMSEs than the other two cases in both temperature and moisture at the mid- and low troposphere (figure not shown). It indicates that the overall simulation of Typhoon Kompasu by the WRF model is not as successful as the other two cases. Nevertheless, the simulated maximum $\Delta\phi$ is about 17 mm near 2.5 km, which agrees with the observation. The large simulated $\Delta\phi$ is contributed to by the snow species. Additionally, most hydrometeors are distributed closer to the western part of the rays.

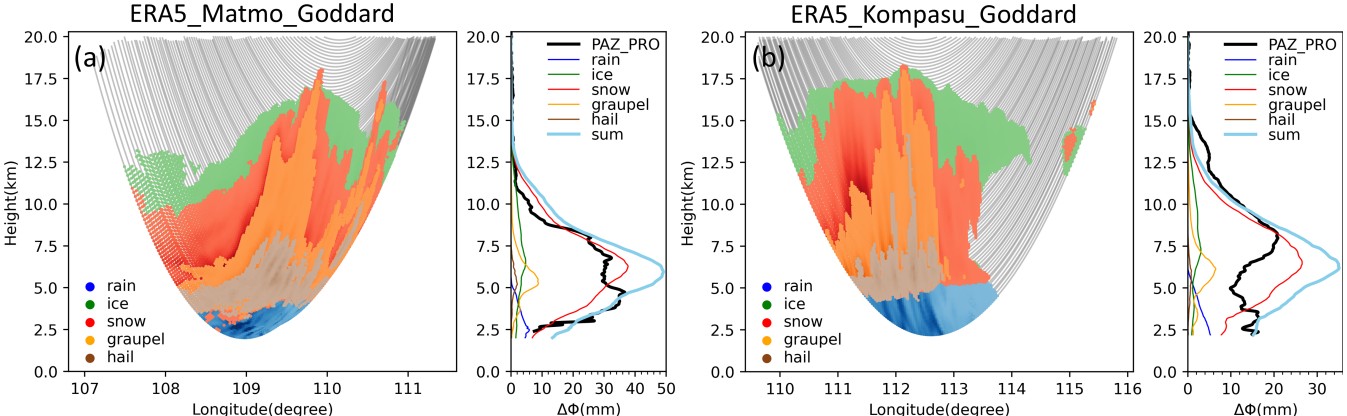

**Figure 8. Distribution of simulated hydrometeors along the PAZ raypath on the left panel, and the differential phase shift from PAZ observation (black curve), along with the calculated phase shift for each simulated hydrometeor species on the right panel. The light blue curve represents the sum of all simulated phase shifts. Panels (a) and (b) show simulations using the Goddard microphysics scheme for Typhoon Matmo and Typhoon Kompasu, respectively.**

To investigate why the $\Delta\phi$ simulations for Typhoon Kompasu do not align well with the PRO, synoptic-scale verifications (against ERA5 analysis) are conducted for three typhoon cases, which are initialized with ERA5 and simulated using the Goddard microphysics scheme (i.e., ERA5_Bualoi_Goddard, ERA5_Matmo_Goddard, and ERA5_Kompasu_Goddard). Fig. 9 presents the verifications in temperature and water vapor mixing ratios for the three simulations. Below 300 hPa, ERA5_Kompasu_Goddard generally exhibits a larger RMSE than the other two cases, particularly for moisture (Fig. 9b), indicating a greater discrepancy of the simulated synoptic environment of ERA5_Kompasu_Goddard from the ERA5 analysis. Discrepancies in the simulated larger-scale environment could lead to deviations in the simulated mesoscale typhoon circulation and associated precipitation, thereby preventing it from capturing the observed $\Delta\phi$ variations.

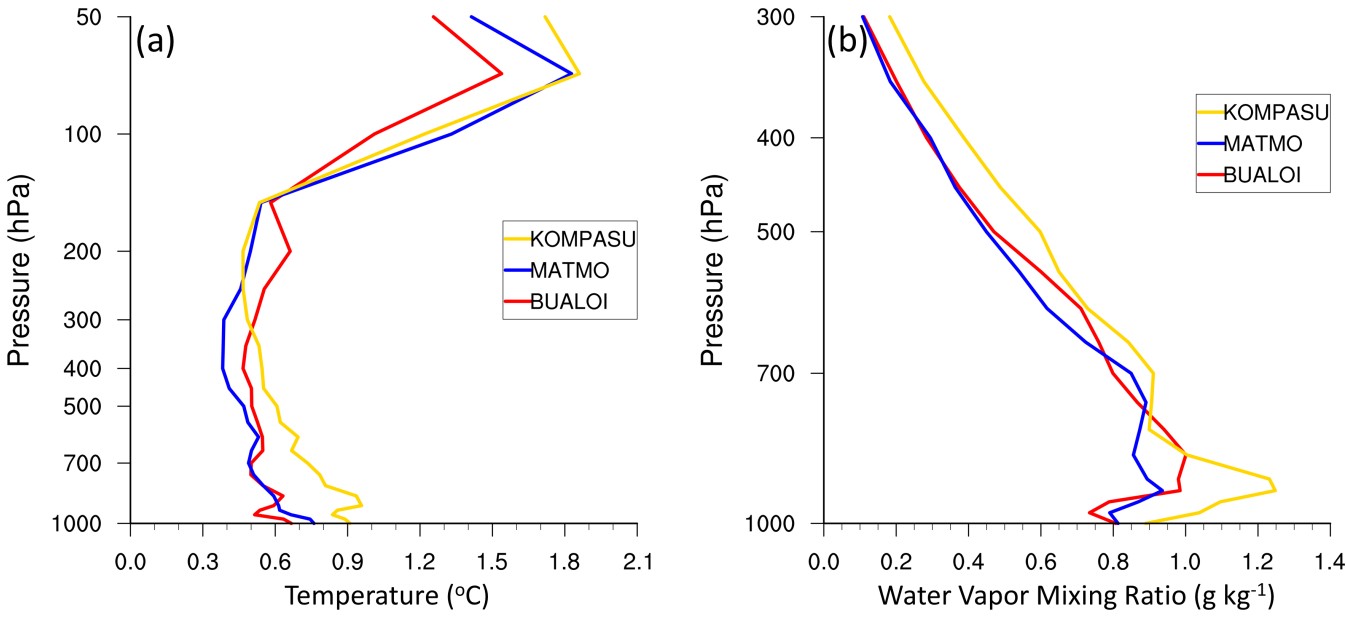

**Figure 9. Averaged grid verification against ERA5 in RMSE for simulation with the Goddard microphysics scheme for (a) temperature and (b) water vapor mixing ratio across all simulations during the 18-hour forecasts. The red curve corresponds to ERA5_Bualoi_Goddard, the blue curve to ERA5_Matmo_Goddard, and yellow curve to ERA5_Kompasu_Goddard.**

### 4.2 Ensemble run with Goddard microphysics

Evaluating hydrometeors within the typhoon circulations can be a challenge because the convective systems surrounding the tropical cyclone is highly variable in time and space. It is difficult to obtain a perfect simulation of an observed convective system surrounding the eye wall at the right time and location. To gain insight on the uncertainties of simulated convective systems, we perform ensemble forecasts for the three typhoon cases. The ERA5_Bualoi_Goddard experiment is adapted as an example. The same initial condition from ERA5 for the deterministic run in Sect. 4.1 was used to create perturbations for 36 ensemble members by using the RANDOMCV function in WRFDA, which made perturbations in control variable space (Barker et al. 2012). Fig. 10 illustrates the variance and ensemble mean for the 36 ensemble initializations at the initial time for Typhoon Bualoi. It shows significant variations in precipitable water over the region (20°N-30°N, 135°E-150°E), indicating large model uncertainties over this area, in the vicinity of the PAZ PRO profile. The same process as the deterministic run, the 36 ensemble members were conducted for 18 h forecasts, and then the PRO data can be used to evaluate ensemble uncertainty.

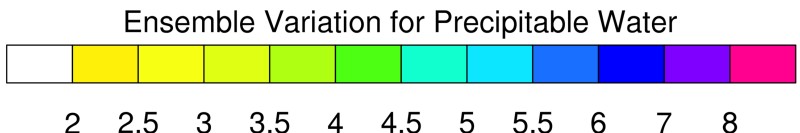

Ensemble Mean for Precipitable Water Contours: 5 to 65 by 5

Ensemble Variation for Precipitable Water

2   2.5   3   3.5   4   4.5   5   5.5   6   7   8

**Figure 10. The ensemble variance (shaded) and ensemble mean (contour) of the precipitable water (unit: mm) for the 36 ensemble members. The cross sign indicates the location of the PRO observation, and the plus sign indicates the best track for Typhoon Bualoi at 0600 UTC 23 October 2019.**

After integrating each of the 36 ensemble members for 15 hours and 40 minutes, the simulation time was closest to the observation time of PRO (i.e., 2019-10-23 21:41:19). The PRO ray path shows a northeast-southwest orientation and passes through the typhoon center (Fig. 11). Fig. 11 shows the radar reflectivity simulation for each member. There are significant variations among the members. Some members, such as member 13, exhibit asymmetric and incomplete eyewall. In addition to slight differences in the typhoon's position, the typhoon intensity and circulation distribution also display significant variations across members. Therefore, a single numerical model simulation may not be representative for the comparison with PRO observations. To capture the uncertainties, we performed an ensemble mean of the 36 simulations and then compared it with the PRO observations.

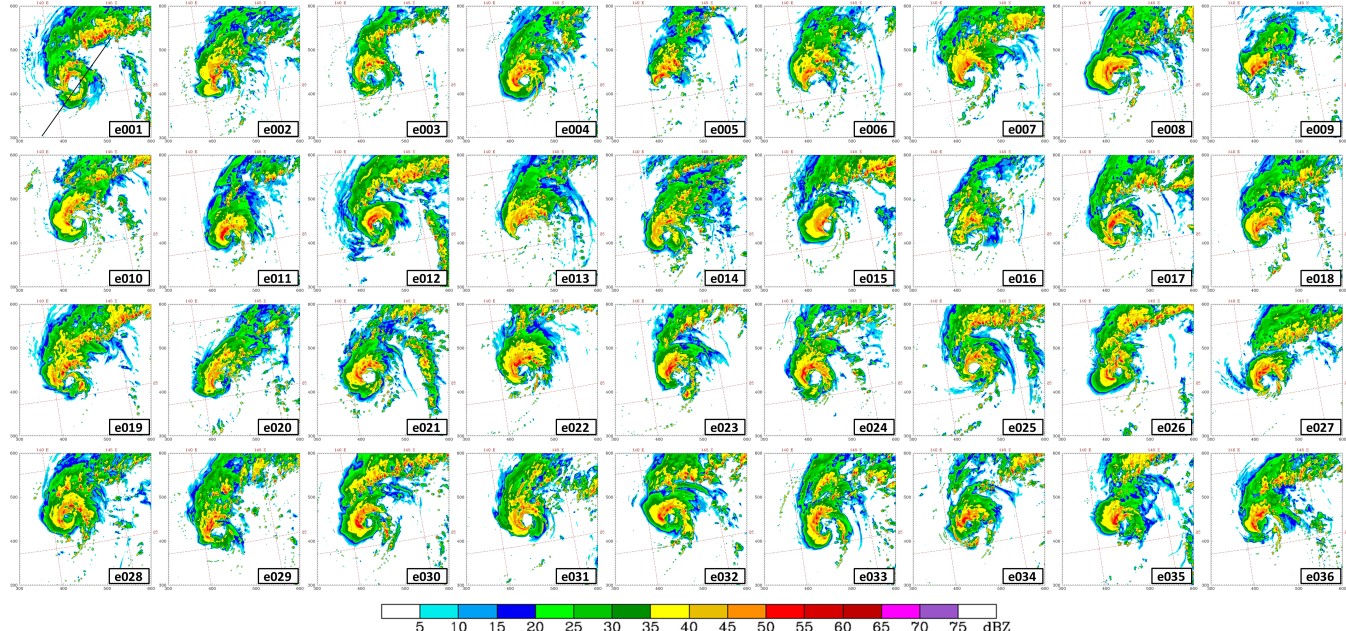

Figure 11. Maximum simulated equivalent radar reflectivity factor for each ensemble member at the 15-hour and 40-minute forecast, compared to the PAZ observation (2019-10-23 21:41:19) for the Bualoi case. The black line in the upper-left panel indicates the PAZ ray path, and the panels, arranged from left to right and top to bottom, follow the member order from 1 to 36.

Because of the difference between the simulated and observed typhoon positions, we considered the vortex position error in each member and adjusted location of the calculated ray path accordingly. After averaging the calculated $\Delta\phi$ from all members, the ensemble mean for the three typhoon cases are shown in Fig. 12. For the Bualoi case, Fig. 12a displays that the ensemble-averaged hydrometeor distribution is more homogenous, and the extreme values are smoothed out, the maximum total $\Delta\phi$ of the ensemble mean is 32.4 mm (Fig. 12a), unlike the more distinct patterns in the single deterministic experiment with a maximum of 35.9 mm (Fig. 6c). Despite this, the pattern of $\Delta\phi_{sum}$ remains consistent with the PRO observations. Fig. 12b shows the variations in the simulated $\Delta\phi_{sum}$ across the 36 members and the ensemble mean for Typhoon Matmo. Within one standard deviation, the ensemble mean can capture the variation of PRO observed profile, showing that the ensemble mean is a good representation of the ensemble. For Typhoon Matmo, the overall result of the ensemble mean is similar to that of the single deterministic forecast (Fig. 12c and 8a), with a maximum $\Delta\phi$ of 43.5 mm, which is approximately 6 mm smaller than the single simulation. The ensemble mean exhibits a smoother vertical variation in the total differential phase shift, but the overall pattern still closely resembles the PRO observations (Figure 12c,d). In contrast, for Typhoon Kompasu, even with 36 ensemble simulations, the simulated phase variations still exhibit significant differences compared to PRO observations. The maximum $\Delta\phi_{sum}$ occurs at around 6 km with a value of 34.4 mm for the ensemble mean, while it is around 21 mm at 8 km from PRO (Fig. 12e). The ensemble results indicate that the simulated phase shift is underestimated above 8 km and overestimated below this level for the Kompasu case (Fig. 12f). This could be related to the performance of the synoptic-scale simulation for Typhoon Kompasu since the background field for the perturbations came from the same source as in the

375 deterministic run. Generally, the variation of the differential polarimetric phase shift from PAZ PRO is comparable to the one standard deviation of the ensemble mean for ERA5_Bualoi_Goddard and ERA5_Matmo_Goddard, which indicates good simulations of the hydrometeors from the numerical model. Consequently, the usefulness of the PRO data for the evaluation of cloud microphysical parameterization is evident.

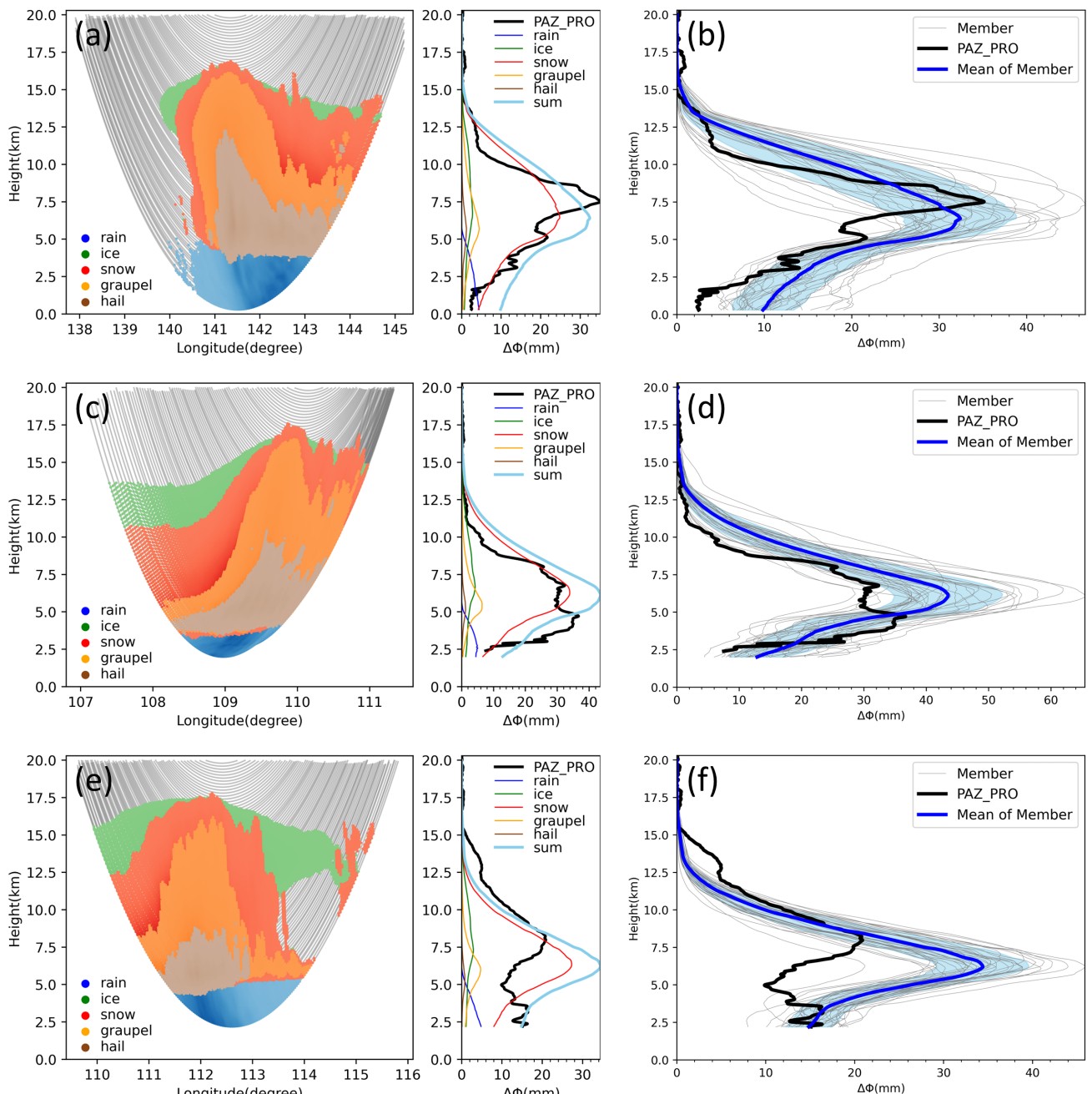

**Figure 12. (a) Same as Figure 6a, but for the ensemble mean and (b) the total differential phase shift for each ensemble member for Typhoon Bualoi. The light blue shadow described the range for one standard deviation. Panels (c),(d) and (e),(f) are the same as (a),(b), but for Typhoon Matmo and Typhoon Kompasu, respectively.**

## 4.3 Uncertainty of planetary boundary layer schemes for ERA5_Bualoi_Goddard

Besides the microphysics parameterization methods that influence precipitation, the planetary boundary layer (PBL)
schemes can make some differences (e.g., Cintineo et al. 2014; Hernández et al. 2024). To assess the uncertainty introduced by PBL parameterizations, we conducted four additional simulations using the ERA5_Bualoi_Goddard dataset but with alternative PBL schemes. While the original configuration used the Yonsei University (YSU) scheme, the new experiments incorporated four other PBL schemes (Table 3): Mellor-Yamada-Janjic (MYJ), Mellor-Yamada Nakanishi and Niino Level 3 (MYNN3), Asymmetric Convective Model version 2 (ACM2), and Grenier-Bretherton-McCaa (GBM).

The changes of the different PBL schemes in the simulated $\Delta\phi$ profiles are shown in Fig. 13. Although these different PBL schemes affect the development of hydrometeors and the simulated precipitation fields, the overall variation is less pronounced than that caused by different microphysics schemes. The variations of the $\Delta\phi$ for each hydrometeor are comparable, and the maximum values of $\Delta\phi$ for YSU, MYJ, MYNN3, ACM2, and GBM PBL schemes are 35.9, 37.2, 27.14, 33.5, and 39.4 mm, respectively. The comparisons of different microphysics and PBL schemes are intended to illustrate
possible sources of model uncertainty, rather than to provide a comprehensive evaluation of all parameterization options. This preliminary comparison demonstrates the value of PRO data in evaluating model performance.

**Table 3. The abbreviated names and planetary boundary layer schemes used and their corresponding WRF options.**

| Abbreviated name | PBL scheme | WRF options |
| --- | --- | --- |
| YSU | Yonsei University scheme | 1 |
| MYJ | Mellor-Yamada-Janjic scheme | 2 |
| MYNN3 | Mellor-Yamada Nakanishi and Niino Level 3 scheme | 6 |
| ACM2 | ACM2 scheme | 7 |
| GBM | Grenier-Bretherton-McCaa scheme | 12 |

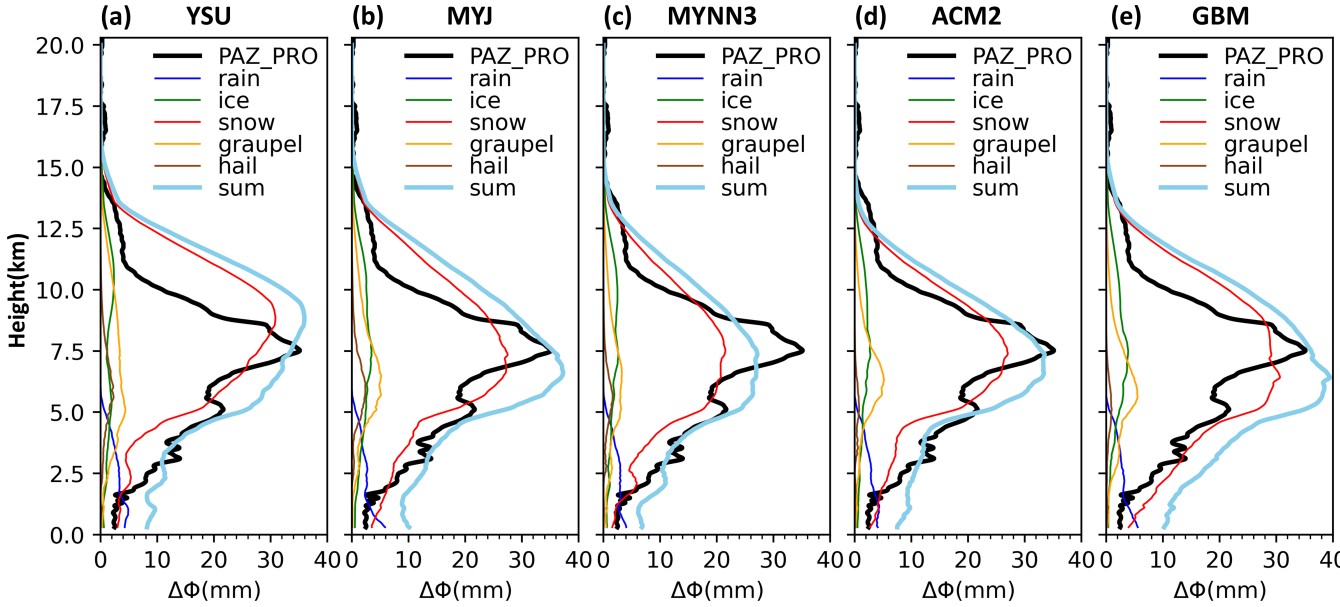

Figure 13. Panel (a) is the same as the right panel of Fig. 6c, using the YSU scheme for PBL parameterization. Panels (b)–(e) are similar to (a), but apply the MYJ, MYNN3, ACM2, and GBM PBL parameterization schemes, respectively.

## 5 Conclusions

The prediction of cloud and precipitation by numerical models is handled by microphysical parameterization schemes, which are commonly developed based on laboratory experiments and empirical data. The evaluation of these microphysical parameterization schemes has been a challenge, because of the lack of observations. By tracking the differential phase shift between horizontal and vertical polarization of GNSS signals as they transverse through precipitation systems, the polarimetric radio occultation (PRO) technique provides information on cloud hydrometeors. Such measurements can potentially be used to evaluate the performance of cloud microphysical parameterization schemes of numerical models.

In this study, we conduct WRF model simulations for three typhoon cases from 2019 and 2021. For each typhoon case, the simulations were conducted with two initial conditions (i.e., ERA5 and NCEP FNL) and five microphysics schemes (PurdueLin, WSM6, Goddard, Thompson, and Morrison). The results show significant variability in the distribution of the simulated hydrometeors, depending on initial conditions, microphysics parameterizations, typhoon location and asymmetric structure, as well as perturbations in the ensemble forecasts. The WRF simulations are interpolated to the location of PAZ PRO observations, taking into consideration of position error of the model tropical cyclones. The simulated cloud hydrometeors are then used to produce the simulated PRO profiles using a forward operator for comparison with the PAZ PRO observations.

The WRF simulations show a similar variation in track predictions for different microphysics schemes, where using the Goddard microphysics scheme predicted a relatively better agreement with observations in terms of both typhoon track prediction and the cloud-top temperature distribution. Most of the microphysics schemes used in the study present a good agreement in the variation of the differential phase verification against the PAZ PRO data, especially for the Goddard, Thompson and Morrison schemes. The superior performance of the Goddard microphysics scheme was supported by both a deterministic forecast as well as the ensemble mean for Typhoon Bualoi and Typhoon Matmo. Furthermore, the PRO observations varied comparable to the one standard deviation range of the ensemble members for both cases, demonstrating the robustness of the verification. Considerable discrepancy between simulated and observed PRO was found for the third case, i.e., Typhoon Kompasu. The verification against ERA5 reanalysis showed that the WRF simulation of Typhoon Kompasu had the largest errors among the three cases.

Even though these preliminary results are encouraging, the limitations of this study should be noted. First, only three PRO profiles for the three typhoon cases were examined. More PRO observations for additional tropical cyclone cases should be evaluated to establish the statistical robustness of the conclusions. Second, there is still a challenge to estimate the physical factors precisely (e.g., turbulence, particle type, and orientation, etc.), that may affect the relationship between the PRO measurement and microphysical assumptions. Moreover, the simulated convective precipitation systems are transient and highly variable, and can vary significantly due to uncertainties in vortex location, large-scale circulations, detailed cloud microphysics, and the model's representativeness of other physical processes (i.e., boundary layer, radiative forcing, etc.). It would be desirable to improve the accuracy of the large-scale and mesoscale simulation of the tropical cyclones prior to the evaluation of the simulated cloud hydrometeors.

With the availability of more PRO data from commercial sources, we plan to expand our study for more typhoon cases in the future. In addition, we will evaluate the performance of cloud microphysics schemes against PRO observation under different weather regimes (e.g., atmospheric river, Mei-Yu front, mesoscale convective systems, etc.).

**Data and code availability**

The initial conditions for all the simulations are downloaded from https://cds.climate.copernicus.eu/api/v2 for ERA5 and https://rda.ucar.edu/datasets/d083003/ for NCEP FNL. The GNSS RO profiles for the validation are downloaded from https://tacc.cwa.gov.tw/data-service/fs7rt_tdpc/level2/wetPf2/. The WRF model is an open-source model, and it can be found at https://github.com/wrf-model/WRF.

## Author contributions

Conceptualization, SYC and YHK; Data Curation and Methodology, SYC, HWL, and RP; Validation, SYC and HWL; Formal Analysis, SYC, YHK, RP, EC, and FJT; Writing - Original Manuscript, SYC; Writing - Review and Editing, SYC, YHK, EC, FJT, HWL, and RP.

## Competing interests

The authors have no competing interests.

## Acknowledgments

This research was supported jointly by Taiwan National Science and Technology Council (NSTC) under Grant NSTC 113-2111-M-008-019 and NSTC 114-2119-M-008-005, and Taiwan Space Agency (TASA) under Grant TASA-S-1130319. The National Center for High-performance Computing (NCHC) is acknowledged for providing computational and storage resources. The efforts of FJT were performed at the Jet Propulsion Laboratory, California Institute of Technology, under a contract with NASA.

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
