# Peer review of "Comparisons of Polarimetric Radio Occultation Measurements with WRF Model Simulation for Tropical Cyclones"

_EGUsphere, 2024_

## Author Comment (AC1)

This study explores the use of polarimetric radio occultation (PRO) measurements for validation and verification of different microphysics schemes implemented on a limited area atmospheric model (WRF). Model simulations of typhoon cases are used to simulate the actually observed PRO measurements using a forward observation operator that is similar to the one developed for ECMWF's IFS model previously reported in the AMT journal. The simulated PRO observations are compared to the actual PRO measurements to gain useful insight into which microphysics scheme performs well in simulating PRO observations. Such comparisons will be potentially very useful given the scarcity of measurements direct related to 3-dimensional distribution of hydrometeor particles, and this study is a nice demonstration of this potential.

One of the major difficulties in such a validation approach would be how to account for erroneous representation of the typhoon position in the model. The authors meticulously accounted for this error source by manually relocating the WRF model field so that the model's typhoon position matches with the position from best track data.

The manuscript is very well organized and written in clear language. Flow of logic is also clear and I see no problem in publishing the manuscript as is except for some minor editorial issues.

I just point out below some minor edits that the authors may find useful, but I do not think these are essential for acceptance of the manuscript.

Thank you very much for the valuable feedback. We greatly appreciate your positive comments on our work. Regarding the minor edits you mentioned, we have carefully reviewed and revised the manuscript accordingly.

Our point-by-point response is provided below in blue for clarity.

Minor comments:

As I understand, when WRF model is initialized, hydrometeor variables are given zero values at the very beginning of the model integration. In such a "cold start" setting for hydrometeor variables, these variables need to be spun-up before any examination is made. It would be informative to readers who may be interested in replicating your

experiments or similar experiments if this point (whether the hydrometeor variables were "cold-started") is explicitly explained in section 2.1.

Thank you for the suggestion. Yes, the experiment was initialized with a cold start. We have added the description: "Each simulation begins with a cold start and is integrated for 18 hours to spin up the model microphysics" in the revised manuscript (line 102) to clarify the initialization process of the hydrometeor variables.

If you did apply cold-start, then I assume the models are integrated for relatively long 18 hours to ensure the model's microphysics is spun-up. If this is the case, this point should also be explained in the manuscript.

Yes, as mentioned in the previous reply, we applied a cold start. To allow sufficient spin-up of the microphysics, the models were integrated for 18 hours for each case. We have clarified this point in the revised manuscript.

Figure 8a: Looking from top to below on the right panel, the observed PAZ data is nearly zero at around 2km height and then rapidly increases as the height gets lower, and this behaviour looks unnatural. I suggest the authors check the quality flag for the PAZ data. If the data is flagged unreliable at these heights, I suggest not to show the PAZ data for such lower levels in the graph. Similarly for Figure 8b and Figure 12.

Thank the reviewer for pointing this out. We have replotted the figures and excluded the flagged data. All figures for PAZ $\Delta\phi$ have been updated, including Figures 6, 8, and 12. The PAZ profile for the Kompasu case is of good quality, and we have removed data below ~2.1 km to maintain consistency with the lowest height of ray tracing.

Typographic issues:

Equation (1) and elsewhere: delta phi should be typed with $\Delta \Phi$" in LaTeX, not with "\Delta \0" or "\Delta \varnothing" as in the manuscript.

We have revised all instances of $\Delta\phi$ in the manuscript to be properly formatted in LaTeX.

Line 175 and elsewhere: "vortexes" should be "vortices".

It has been corrected.

Line 177 "Even though": should be replaced with "Despite", or the sentence structure should be revised.

It has been revised accordingly.

Lines 197, 249 etc.    "presented": should be "present"

It is done.

Line 198 "five schemes however, ...": Start a new sentence with "However", like "...five schemes. However, ..."

It is done.

Line 269 "-70 degrees": Make it clear that this is Celsius.

It has been revised to -70ºC.

Line 296 "contributed by": Probably should be "contributed to by".

It is done.

---

## Author Comment (AC2)

We thank the reviewer for the valuable and insightful comments. In response, we have revised the manuscript to extend discussions on the potential challenges and limitations of the approach. As suggested, we also conducted additional simulations using different PBL schemes for case ERA5_Bualoi_Goddard to assess their influence further. Our point-by-point responses are provided below in blue.

The manuscript explores the impact of several microphysics schemes on the polarimetric signature, during radio occultation with polarimetric-capable receivers. The paper shows that the different schemes lead to different expected observables. This difference is clearly above noise for the observable, thus these observations can in principle support the superiority of some microphysical schemes above others.

The theoretical basis for this is in general appropriate, and the authors demonstrate what I understand is the main goal, which is to show that the polarization signature is measurable with better accuracy than the difference between microphysical schemes. Indeed, some schemes lead to significantly better fits than alternative microphysics. This is interesting. Despite this, the authors do not explore sufficiently the caveats of the approach. The relationships between water precipitates and polarization signatures depend on the amount of water/ice, and the average axis ratio. Although the amount of water/ice is quite explicit in any microphysics scheme, the effective axis ratio is hardly an output of any standard scheme. It is here somewhat arbitrarily fixed to a very crude guess of 0.5, and it is unclear how other choices of this quantity may impact the results. It may be a different constant, a profile dependent on the type of precipitate, and depends likely also on turbulence.

Thank you to the reviewer for this insightful comment. We agree that the relationship between cloud hydrometeors and polarization signatures is influenced by multiple complex factors, including the amounts of hydrometeors and the effective axis ratio (*ar*), which is determined by the particle size distribution and orientation. Both factors are difficult to constrain accurately, and this introduces a limitation to our approach.

In our study, we did not assume a constant axis ratio as in some previous works. Instead, we applied a height-dependent profile of $\rho \cdot (1-ar)$, which is based on Fig. 9 of Padullés et al. (2022). This profile assumes a fixed density of $\rho = 0.2$ *g m*$^{-3}$ and a variable axis ratio that changes with temperature. The temperature dependence is supported by satellite observations on frozen hydrometeors, as discussed in Padullés et al. (2022). Beyond the defined temperature range (e.g., above the cloud top or below the freezing

level), $\rho \cdot (1-ar)$ is held constant at its value at the nearest boundary level (Fig. R2-1a, black line).

In the original manuscript, the same $\rho \cdot (1-ar)$ function was applied to all the hydrometeors, as represented by the black line in Fig. R2-1a. However, since this function was estimated for frozen hydrometeors, we refer to Chang et al. (2009), which suggests an axis ratio of $ar = 0.95$ for liquid rain during typhoon events (with a density of $\rho=1$ g cm$^{-3}$). Therefore, a fixed value of $\rho \cdot (1-ar) = 0.05$ is used for rain, as shown by the blue line in Fig. R2-1a. The revised manuscript has been updated accordingly, using different $\rho \cdot (1-ar)$ functions for solid and liquid hydrometeors. This methodology has been clarified in the revised manuscript (Section 2.2).

To further address the concern regarding the fixed axis ratio, we conducted sensitivity tests by applying various constant $ar$ values (ranging from 0.1 to 0.9) for solid hydrometeors, and a fixed $ar = 0.95$ for rain, in the ERA5_Bualoi_Goddard case. The results (shown in Fig. R2-1b) reveal that while the general shape of the $\Delta\phi$ curves remain similar, the maximum $\Delta\phi$ tends to occur at higher altitudes. The curve used in the revised manuscript (i.e., the "curve fit") is closer to the PRO observation, indicating a relatively better representation for $\Delta\phi$ calculation. It is also noted that larger $ar$ values tend to produce smaller $\Delta\phi$.

In the revised methodology for $\Delta\phi$ estimation, hydrometeor phase (solid or liquid) is taken into account when computing the simulated phase shift. To mitigate potential representativeness errors arising from rainband variability, $\Delta\phi$ values are averaged along the relocated raypath as well as two parallel paths offset by 0.5°. The associated figures presenting $\Delta\phi$ have been updated in the revised manuscript. In addition, we have added a brief discussion in the revised manuscript regarding potential sources of uncertainty and physical factors (e.g., turbulence, particle type, and orientation) that may affect the relationship between PRO measurements and microphysical assumptions.

[Figure]

Fig. R2-1. (a) The $\rho \cdot (1-ar)$ function used for solid (black line) and liquid (blue line) hydrometeors. (b) Sensitivity test of $\Delta\phi$ profiles for case ERA5_Bualoi_Goddard, using fixed axis ratio ($ar$) values ranging from 0.1 to 0.9 for solid hydrometeors. The black line represents the PRO observation, and the light blue line shows the curve fitting adopted in the revised manuscript. Other colored lines correspond to different fixed $ar$ values, as indicated in the legend.

Besides, the microphysics interact, as is mentioned in the paper, with PBL schemes. Given this wide parameter space, above the mere amount of several precipitate fractions, it is not obvious that we could at this point conclude that some microphysics scheme is superior based on PRO data. We can conclude, however, that through its accuracy and resolution, PRO data has the ability to discern different schemes. It is my understanding that we still ignore too much of the microphysics and of other related parameterizations, such as PBL schemes, to actually benefit from that ability, even if PRO data is available.

We agree that several parameters, including planetary boundary layer (PBL) schemes, can influence precipitation. To assess the uncertainty introduced by PBL parameterizations, we conducted four additional simulations using the ERA5_Bualoi_Goddard setup, but with alternative PBL schemes. While the original configuration used the Yonsei University (YSU) scheme, the new experiments incorporated four other PBL schemes (Table R2-1): Mellor-Yamada-Janjic (MYJ), Mellor-Yamada Nakanishi and Niino Level 3 (MYNN3), Asymmetric Convective Model version 2 (ACM2), and Grenier-Bretherton-McCaa (GBM). Although these different PBL schemes affect the development of hydrometeors and the simulated

precipitation fields, the overall variation is less pronounced than that caused by different microphysics schemes, as shown in Fig. R2-2.

Nevertheless, the primary objective of our study is to assess the potential of PRO observations. The comparisons of different microphysics and PBL schemes are intended to illustrate possible sources of model uncertainty, rather than to provide a comprehensive evaluation of all parameterization options. This preliminary comparison demonstrates the value of PRO data in evaluating model performance. Accordingly, we have revised the manuscript to mention these sources of uncertainty and have softened the conclusions to reflect these findings.

Table R2-1.   The abbreviated names and planetary boundary layer schemes used and their corresponding WRF options.

| Abbreviated name | PBL scheme | WRF options |
|---|---|---|
| YSU | Yonsei University scheme | 1 |
| MYJ | Mellor-Yamada-Janjic scheme | 2 |
| MYNN3 | Mellor-Yamada Nakanishi and Niino Level 3 scheme | 6 |
| ACM2 | ACM2 scheme | 7 |
| GBM | Grenier-Bretherton-McCaa scheme | 12 |

[Figure]

Fig. R2-2. Sensitivity of simulated $\Delta\phi$ profiles to different planetary boundary layer (PBL) schemes using the ERA5_Bualoi_Goddard configuration. The panels show simulations using (from left to right) the YSU, MYJ, MYNN3, ACM2, and GBM PBL schemes. The black line indicates the observed $\Delta\phi$ from PAZ PRO. Colored lines represent the contributions from individual hydrometeor types: rain (blue), ice (green), snow (red), graupel (orange), and hail (brown). The light blue line indicates the sum of all hydrometeor contributions.

I thus encourage the authors to underscore the difficulties that would limit the task of supporting a scheme as unconditionally superior to others, based on PRO, and further develop the caveats of the approach. I believe that this can be done with an appropriately

extended comments and conclusion section (beyond the few comments in lines 393-etc).

We agree with the reviewer and sincerely thank you for the valuable suggestion. We have revised the manuscript to better highlight the limitations of our approach. The revisions aim to clarify the scope and constraints of our findings, and to emphasize the importance of considering interactions with other parameterizations.

---

## Author Comment (AC3)

**Reviewer 3**

**Review of "Comparisons between Polarimetric Radio Occultation Measurements with WRF Model Simulation for Tropical Cyclones"**

This manuscript presents a comparative study of three typhoon cases using five different microphysics schemes and two initial conditions (ICs), validated against polarimetric radio occultation (PRO), conventional radio occultation (RO) retrievals, and radiosonde data. The results highlight the unique capabilities of PRO in capturing hydrometeor structures, offering valuable insights for evaluating microphysics schemes.

Overall, the paper is well written and clearly structured. The language is easy to follow, and there are no major grammatical issues. The topic is timely and relevant, and the use of PRO for model evaluation is an important and innovative direction.

However, I have several concerns that need to be addressed before the paper can be considered for publication:

We thank the reviewer for the valuable feedback. We have carefully addressed all concerns and revised the manuscript accordingly. Detailed responses are provided below in blue. We appreciate your thoughtful review.

**1. Use and Interpretation of Initial Conditions (ICs):**

The manuscript does not clearly articulate the purpose and implications of the two different initial conditions. While it is acknowledged that ICs can influence microphysics scheme performance, the two ICs used in this study (ERA5 and FNL) are significantly different even before considering the influence of microphysics as stated in the paper. The author used ERA5 to verify the results but didn't provide independent verification of typhoon intensity and track (e.g., best track data or satellite-derived observations), therefore it is difficult to assess which combination of IC and microphysics performs better against observations. Besides, ERA5 is a reanalysis product that is not truly independent of the model, which weakens its role as a verification dataset. In its current form, the IC study part lacks a clear scientific objective or demonstrated relevance to the overall goals of the study, and its inclusion is difficult to justify without more though validation.

We appreciate the reviewer's comment regarding the role of initial conditions (ICs) in this study. Both ERA5 and NCEP FNL datasets were used as ICs to assess whether uncertainties in the microphysics schemes persist across different initial conditions. NCEP FNL is a widely used IC dataset for regional weather models. ERA5, on the other hand, assimilates a large amount of observational data and is regarded as one of the

most accurate estimates of the atmospheric state (e.g., Beck et al., 2022; Sheridan et al., 2020). It also serves as an alternative option for evaluating the uncertainty associated with microphysics schemes. Additional descriptions clarifying the purpose of using two different initial conditions have been added to the revised manuscript.

Regarding the independent verification, we conducted comparisons between the WRF simulations and the infrared satellite imagery, as shown in Fig. R3-1. A similar comparison for the ERA5_Bualoi_Goddard simulation was presented in Fig. 7 of the original manuscript. Among the five simulations using different microphysics schemes, ERA5_Bualoi_Goddard exhibits a pattern most consistent with the satellite observations (Fig. R3-1). In contrast, the patterns produced by the other schemes appear either too weak and disorganized (e.g., Purdue Lin, WSM6, Thompson) or overly intense and exaggerated (e.g., Morrison), showing less resemblance to the satellite observations. Fig. 7 in the original manuscript has been replaced with Fig. R3-1 to provide a more comprehensive comparison.

[Figure]

Fig. R3-1. The upper panels from left to right are the infrared satellite image with NHC enhancement, adopted from the Cooperative Institute for Meteorological Satellite Studies/University of Wisconsin-Madison (CIMSS), for Typhoon Bualoi at 2230 UTC on 23 Oct. 2019; the 16-h forecast (i.e., 2200 UTC 23 Oct.) cloud top temperature for typhoon Bualoi with the Purdue_Lin and WSM6 microphysics and the initial condition from ERA5. The bottom panels from left to right are the same but for the cloud top temperature with the Goddard, Thompson, and Morrison microphysics.

In addition, we include the simulations of typhoon tracks for all simulation cases (Fig. R3-2) to address the reviewer's concern regarding verification. Under the same initial condition (either NCEP FNL or ERA5), the simulated typhoon tracks using different microphysics schemes exhibit high similarity, indicating a relatively weak

sensitivity of track prediction to microphysics schemes. However, the use of different initial conditions leads to a clear bifurcation in the track patterns, forming two distinct groups corresponding to the two IC datasets. An exception is found in ERA5_Matmo_WSM6, which shows a significantly deviated track compared to the others (Fig. R3-2b).

The analysis of track and intensity errors reveals that most simulated tracks exhibit errors of less than 100 km during the 18-hour forecasts, and the simulated sea-level pressure generally deviates by less than 12 hPa, except for the Bualoi cases. Typhoon Bualoi exhibited stronger development and had a maximum intensity of 935 hPa at the initial time. The simulations with ERA5 IC have a weak intensity at the beginning, resulting in larger intensity errors across all experiments (Fig. R3-3b). To average across the three typhoon cases, Fig. R3-4 presents the mean track and intensity errors. Among all the microphysics schemes, the Goddard scheme generally yields smaller track errors, regardless of the initial condition used (Fig. R3-4a). However, it shows a larger mean intensity error in simulations initialized with ERA5 (Fig. R3-4b). Overall, no consistent pattern emerges to indicate that one IC-microphysics combination outperforms others across all metrics. Therefore, it is difficult to isolate which factor (initial condition or microphysics) has the greatest impact solely based on track or intensity verification.

[Figure]

Fig. R3-2. Simulated tracks over time for Typhoons (a) Bualoi, (b) Matmo, and (c) Kompasu. The best track from JTWC is shown as a black line. Solid lines represent simulations with ERA5 initial conditions, while dashed lines represent those with NCEP FNL initial conditions. Lines in different colors represent different microphysics schemes.

[Figure]

Fig. R3-3. Simulated (a) track and (b) intensity errors over time for Typhoon Bualoi. The best track data are from JTWC. Solid lines represent experiments with ERA5 initial conditions, while dashed lines represent those with NCEP FNL initial conditions. Lines in different colors represent different microphysics schemes. Panels (c) and (d), and (e) and (f) show the same variables as (a) and (b), but for Typhoon Matmo and Typhoon Kompasu, respectively.

[Figure]

Fig. R3-4. The averaged (a) track errors and (b) intensity errors, respectively, from all three typhoon cases. Solid lines represent simulations with ERA5 initial conditions, while dashed lines represent those with NCEP FNL initial conditions. Lines in different colors represent different microphysics schemes.

**Reference**:

Beck, H. E., A. I. J. M. van Dijk, P. R. Larraondo, T. R. McVicar, M. Pan, E. Dutra, and D. G. Miralles, 2022: MSWX: Global 3-Hourly 0.1° Bias-Corrected

Meteorological Data Including Near-Real-Time Updates and Forecast Ensembles. *Bull. Amer. Meteor. Soc.*, **103**, E710–E732, https://doi.org/10.1175/BAMS-D-21-0145.1.

Sheridan, S. C., C. C. Lee, and E. T. Smith, 2020: A comparison between station observations and reanalysis data in the identification of extreme temperature events. *Geophys. Res. Lett.*, **47**, e2020GL088120, http://doi.org/10.1029/2020GL088120.

**2. Use of RO Retrievals as Verification:**

The study treats RO temperature and moisture retrievals similarly to radiosonde data. However, this assumption is problematic for several reasons:

- RO retrievals are often derived using a priori information from climatology or model fields, which compromises their use as independent observational data.

- RO measurements are inherently integrated along ray paths, similar to PRO, and thus are not directly equivalent to point measurements like radiosondes.

- The comparison would be more meaningful if the model outputs were validated against rawer RO observables, such as bending angle. Although these raw RO quantities may not provide hydrometeor information like PRO, they are more suitable for comparing the thermodynamic structure along the ray path and could complement the PRO analysis. This important aspect seems to be overlooked, limiting the insightfulness of the comparisons shown in Section 3.

Thank you for this insightful comment. We agree that RO temperature and moisture retrievals are not equivalent to radiosonde measurements, as they incorporate a priori information, which may compromise their independence. However, the quality of RO retrievals has been extensively validated in previous studies, demonstrating strong consistency with radiosonde observations, particularly for temperature (e.g., Wee et al., 2020; Zhang et al., 2024). Therefore, we consider it acceptable to use RO retrievals as a reference for model verification in certain contexts.

In response to the reviewer's concern, we have conducted a complementary comparison using RO refractivity, as shown in Fig. R3-5. While the RO bending angle is a more fundamental measurement than refractivity, deriving simulated bending angles requires extrapolation beyond the model top, which may introduce additional uncertainties. To avoid this issue, we apply a local refractivity forward operator to compute model refractivity, which can be directly compared with observed RO refractivity profiles. The results shown in Fig. R3-5 exhibit a variation pattern consistent with that in water vapor mixing ratio (Fig. 5b). We believe this enhances the robustness of our model

evaluation by providing a more appropriate comparison with raw RO observables. The description of RO refractivity verification has been added to the revised manuscript.

[Figure]

Fig. R3-5. The mean differences for verifications against RO refractivity across all simulations. The red curve represents the Purdue_Lin scheme, blue represents WSM6, yellow represents Goddard, purple represents Thompson, and green represents the Morrison scheme.

**Reference**:

Wee, T. K., R. A. Anthes, D. C.Hunt, W. S. Schreiner, and Y. H. Kuo, 2022: Atmospheric GNSS RO 1D-Var in use at UCAR: Description and validation. *Remote Sensing*, **14**(21), 5614. https://doi.org/10.3390/rs14215614

Zhang, Z., T. Xu, N, Wang, F. Gao, S. Li, and L. Bastos, 2024: Evaluation of the retrieved temperature and specific humidity from COSMIC-2 and FY-3D with radiosonde, reanalysis data, and MetOp. *Measurement*, **235**, 114936. https://doi.org/10.1016/j.measurement.2024.114936

**3. PRO Verification and Typhoon Structure Alignment:**

The use of PRO for typhoon evaluation is promising, but the manuscript does not sufficiently address potential misalignment between observed and simulated typhoon structures. While center relocation is applied to correct for gross displacement, finer structural differences—such as asymmetries in precipitation bands or peripheral wind fields—are crucial for accurate PRO comparisons. Unlike atmospheric rivers (ARs), typhoons are highly sensitive to orientation and mesoscale features. If the PRO ray trajectory does not intersect the simulated hydrometeor regions properly, it can lead to misleading differences, regardless of microphysics scheme performance.

For example, in Figure 6, Purdue Lin and WSM6 show narrower hydrometeor regions

compared to the other schemes. Is this due to genuine model differences, or a result of misalignment between PRO ray paths and the simulated storm structures? The paper would greatly benefit from incorporating additional observational data (e.g., precipitation from satellite sensors) to independently verify which simulations better match reality. This would help determine whether PRO is truly capturing physical differences among the MP schemes, or whether spatial mismatches are driving the observed discrepancies.

We agree with the reviewer that center relocation alone may not fully resolve the spatial misalignment between observed and simulated typhoon structures. As shown in Fig. R3-1, the cloud structures of different microphysical schemes may differ significantly. To address this issue and better account for fine-scale structural discrepancies, we have implemented an enhanced approach in the revised manuscript. Specifically, in addition to the primary relocated ray path, we include two supplementary ray paths offset by ±0.5 degrees (depending on the ray orientation). We then compute the phase shift delay ($\Delta\phi$) along all three ray paths and use the averaged value as the representative $\Delta\phi$ for comparison with the PRO observations.

This approach helps mitigate the effects of minor misalignments between the observed and simulated storm structures, particularly in highly asymmetric systems such as typhoons. Accordingly, Figures 6, 8, and 12 in the revised manuscript have been updated to reflect the averaged $\Delta\phi$ values derived from this three-ray-path method. We believe this modification enhances the robustness of the PRO-based verification and provides a more reliable evaluation of microphysics scheme performance.

**Conclusion:**
This study has strong potential to make a significant contribution to the field. The use of PRO for evaluating typhoon simulations is novel and valuable. However, the current manuscript does not fully address the limitations of its verification strategy, particularly with respect to the IC interpretation, RO data usage, and structural alignment in PRO comparisons. Addressing these issues would strengthen the scientific rigor and impact of the work.

I encourage the authors to further develop the analysis, especially around item (3), and to consider incorporating additional independent observations to support the conclusions.

We sincerely thank the reviewer for the encouraging remarks and constructive suggestions. We appreciate the recognition of the novelty and potential value of using PRO for typhoon simulation evaluation. In response to the reviewer's concerns, as

mentioned in the replies above, we have carefully revised the manuscript to improve the scientific rigor of our verification strategy. We hope these revisions and clarifications adequately address the reviewer's concerns and strengthen the contribution of this study.

---

## Author Response (AR2)

**Review of "Comparisons of Polarimetric Radio Occultation Measurements with WRF Model Simulation for Tropical Cyclones"**

1. The revised figures, along with the additional verification of track and intensity, are valuable and effectively address my concerns. Thank you for making these improvements.

2. The authors used refractivity to verify the moisture results. However, refractivity here is still treated as a point value under the assumption of being spherically symmetric, allowing it to be calculated at vertical levels rather than along the actual ray trajectory. Ideally, bending angle would be the more appropriate variable for this purpose. I understand, however, that the WRF model lacks a bending angle forward operator, which would make such an analysis beyond the scope of this paper. Given that limitation, the current approach is acceptable.

3. I particularly like the approach taken in the revised Fig. 6, which accounts for the misalignment between the typhoon simulation and the PRO trajectory. However, in L357, the phrase "minor misalignments" seems to downplay the magnitude of the differences (I understand the authors probably use minor here for the distance though). As shown in the comparison between the original and revised Fig. 6, the misalignment between the typhoon's asymmetric structure and the PRO trajectory can be quite significant, with substantial changes in the Ice and total values for most schemes. I suggest revising the wording to avoid implying that the misalignments are not significant, as this could misrepresent their potential impact on interpretation. Anyway, this clearly indicates that the misalignment between the typhoon's asymmetric structure and the PRO trajectory can have significant impacts on the result interpretation, and if not considered appropriately, it could lead to incorrect conclusions. It also reveals a notable limitation of current model simulations and the use of PRO information in such contexts: if the model's storm center location and asymmetric structure are not both well aligned with reality, assimilation of PRO data could either introduce instability/negative impacts or have no impact at all. While data assimilation is not the focus of this study, these results actually highlight a powerful alternative use of PRO: as a validation dataset to diagnose the degree of misalignment between model simulations and observations. This might worthy of authors to emphasize as the value of PRO.

We sincerely thank the reviewer for the constructive and detailed comments. Following the suggestions, we have revised the manuscript accordingly. Specifically, we removed the word "minor" when describing the misalignments and emphasized that the

simulated typhoon's asymmetric structure may deviate from reality and should be considered. We believe these revisions more accurately reflect the potential impact of the misalignment between the typhoon simulation and the PRO trajectory.